# Adaptive dynamics of extrachromosomal circular DNA in rice under nutrient stress

Hanfang Ni [1,2,5], Lenin Yong-Villalobos [3,5], Mian Gu[1,2], Damar Lizbeth López-Arredondo [3], Min Chen[1,2], Liyan Geng[1,2], Guohua Xu [1,2] ✉ & Luis Rafael Herrera-Estrella [3,4] ✉

Extrachromosomal circular DNAs (eccDNAs) have been identified in various eukaryotic organisms and are known to play crucial roles in genomic plasticity. However, in crop plants, the role of eccDNAs in responses to environmental cues, particularly nutritional stresses, remains unexplored. Rice (*Oryza sativa* ssp. *japonica*), a vital crop for over half the world's population and an excellent model plant for genomic studies, faces numerous environmental challenges during growth. Therefore, we conduct comprehensive studies investigating the distribution, sequence, and potential responses of rice eccDNAs to nutritional stresses. We describe the changes in the eccDNA landscape at various developmental stages of rice in optimal growth. We also identify eccDNAs overlapping with genes (*ecGenes*), transposable elements (*ecTEs*), and full-length repeat units (*full-length ecRepeatUnits*), whose prevalence responds to nitrogen (N) and phosphorus (P) deficiency. We analyze multiple-fragment eccDNAs and propose a potential TE-mediated homologous recombination mechanism as the origin of rice's multiple-fragment eccDNAs. We provide evidence for the role of eccDNAs in the rice genome plasticity under nutritional stresses and underscore the significance of their abundance and specificity.

Exploring genomic plasticity has led to the identification of extrachromosomal circular DNAs (eccDNAs) across a wide range of eukaryotes[1]. Using light microscopy, large extrachromosomal circular molecules known as Double Minutes (DMs) were initially observed in mammalian cells and higher plants[2]. These observations supported Franklin Stahl's hypothesis that DNAs might be organized as circular molecules within chromosomes in higher organisms[3]. Subsequent analyzes revealed that DMs carried chromatin bodies with chromosomal homologous sequences of megabase length but lacking telomeres and centromeres, linking them to oncogene amplification in tumors in mammals[4]. Further research highlighted the presence of eccDNAs in diverse eukaryotic cells, including yeast (*Saccharomyces cerevisiae*) and other higher plants[5-7]. Moreover, the chromosomal topology rearrangements caused by the formation of large eccDNAs can co-amplify enhancer elements and over-express oncogenes[8,9]. Smaller eccDNAs, which cover only exon fragments from genes, can produce mature regulatory short RNAs and modulate the expression of their chromosomal counterparts[10]. Additionally, eccDNAs have been shown to possess innate immunostimulatory activity, primarily due to their circularity, independent of their sequence content[11].

In yeast, eccDNAs harboring functional genes enhance environmental adaptability and drive genomic evolution. For instance, in

[1]National Key Laboratory of Crop Genetics & Germplasm Enhancement and Utilization, Nanjing Agricultural University, Nanjing, China. [2]MOA Key Laboratory of Plant Nutrition and Fertilization in Lower-Middle Reaches of the Yangtze River, Nanjing, China. [3]Department of Plant and Soil Science, Institute of Genomics for Crop Abiotic Stress Tolerance (IGCAST), Texas Tech University, Lubbock, TX, USA. [4]Unidad de Genómica Avanzada/Langebio, Centro de Investigación y de Estudios Avanzados del Instituto Politécnico Nacional, Irapuato, Gto. Mexico. [5]These authors contributed equally: Hanfang Ni, Lenin Yong-Villalobos. ✉e-mail: ghxu@njau.edu.cn; luis.herrera@ttu.edu

yeast, the copy number of the copper-resistance gene *CUP1* increases via eccDNA formation in response to environmental stress[12]. Likewise, limiting nitrogen (N) supply enhances the expression levels of the general amino acid permease gene *GAP1* in yeast by eccDNA amplification (*GAP1circles*), facilitating the transport of amino acids across the plasma membrane[13]. *GAP1circle* are produced via Homologous Recombination (HR) between two Long Terminal Repeat (LTR) elements surrounding the chromosomal *GAP1* gene, causing deletions and supporting the role of HR in eccDNA formation[13]. These studies highlight the crucial role of eccDNAs in environmental stress responses in eukaryotic cells.

The presence of eccDNAs has been reported in several plant species[14], including Arabidopsis (*Arabidopsis thaliana*)[15], sugarcane (*Saccharum officinarum*)[16], rice (*Oryza sativa*)[17], and Palmer amaranth (*Amaranthus palmeri*)[18]. The latter represents a notable example of the functional role of eccDNAs in plants conferring glyphosate resistance[18] in the noxious weed *Palmer amaranth*, resulting from the eccDNA increased copy number of the gene encoding the target enzyme of this herbicide, 5-enolpyruvylshikimate-3-phosphate synthase (EPSPS)[19]. This eccDNA replicon, which carries *EPSPS* along with 58 other genes, contributes to widespread glyphosate resistance in *Amaranthus palmeri* across America[20,21]. Additionally, bioinformatic analysis of eccDNAs in cold-tolerant potato cultivars identified valuable markers for molecular breeding[22]. These findings suggest that eccDNAs in plants play a crucial role in rapid adaptation and evolution, highlighting their potential impact on plant biology and agricultural practices.

Analyzing eccDNA sequences can also aid in identifying active LTRs in different plant tissues and enhance our understanding of Transposable Elements (TE)-driven evolution and genetic adaptation[22,23]. High TE-derived eccDNA loads in Arabidopsis mutants of the chromatin remodeler DDM1(Decreased DNA methylation1), *ddm1*, suggested a role for eccDNAs in genome instability by altering DNA repair pathways[23]. Recently, dynamic analysis in rice also revealed significant insights on eccDNA origin[17]. Although increasing data on the functional roles of eccDNAs on adaptive mechanisms and genome evolution has been recently reported, the role of eccDNAs in plant adaptation to environmental pressures, including nutritional stresses, remains largely unexplored[7].

To address this gap, we investigate the landscape of both single-fragment and multiple-fragment eccDNAs in rice plants subjected to nitrogen (N) and phosphorus (P) stresses. We examine the differences in the abundance of genes and TEs in rice eccDNAs during optimal growth and in response to N and P stresses. Our results demonstrate that eccDNAs respond to fluctuations in nutritional supply and contribute to rice genomic plasticity.

## Results

### eccDNAs are derived from diverse regions across the rice genome

Rice (*Oryza sativa* ssp. *japonica* cv. *Nipponbare*) shoot tissues were collected from eight treatments with two biological replicates each to isolate eccDNAs. For the reliable observation of nutrient limitations during different growth stages, strict optimal nutrient growing conditions were set for 1 (*Ctrl_D1*), 3 (*Ctrl_D3*), 7 (*Ctrl_D7*), and 14 days (*Ctrl_D14*), and low nitrogen (LN) treatments set for 3 (*LN_D3*) and 7 days (*LN_D7*), low phosphorus (LP) treatments for 7 (*LP_D7*) and 14 days (*LP_D14*) (Supplementary Figs. 1a–b; see "Method"). Shoot tissue was selected for this analysis as the most severe symptoms of nutrient deficiency are observed in leaves, and a higher eccDNA accumulation has been reported in the shoot compared to the root[17]. After removing contaminant chromosomal DNA, the extracted eccDNAs were purified and then amplified using random Rolling Circle Amplification (rRCA) to generate high-quality libraries for Oxford Nanopore long-read sequencing (Supplementary Fig. 1c, see "Method"). Arbitrarily primed-like PCR (apLike PCR) was used to

confirm the total removal of linear DNA and *Eco*R1 digestion of rRCA products to confirm the isolation and amplification of eccDNAs (Supplementary Fig. 2a–b). After de-branching on amplified eccDNAs, high-quality samples were used for Oxford Nanopore sequencing libraries. Nanopore sequencing data was further quality controlled (Supplementary Data 1) and analyzed as described in the "Methods" section and Supplementary Fig. 3a. Long reads facilitate eccDNA characterization compared to Illumina short-read sequencing, which is generally insufficient to identify eccDNAs from repeated loci of large genomes. Instead of using the advanced tools of CIDER-Seq2[24], FLED[25,] and CReSIL[26], we opted for ecc_finder[27] for eccDNA identification in rice. This decision was based on ecc_finder's precise bioinformatics framework, which accounts for genomic repeats, applies size filtering, and optimizes redundancy for eccDNA characterization (Supplementary Fig. 3b). Moreover, ecc_finder has performed excellently with long-read Nanopore sequencing data in plants[17,23,27].

Only reads mapping to the rice chromosomal genome were kept during the eccDNA identification through the long-read pipeline of ecc_finder. Analysis showed that the eccDNA size distribution ranged from 200 bp to 37 kb, with all eccDNAs having a minimum size of 200 bp due to ecc_finder's identification criteria (Figs. 1a and Supplementary Fig. 3b). Although a few eccDNAs were over 37 kb, the average size across all the treatments was ~500 bp. This size distribution was similar to that observed in other organisms, such as human cells[11], Arabidopsis[15], and the short-read sequencing analysis from rice[17] (Supplementary Fig. 4a). It is worth mentioning that there were no significant statistical differences in size distribution between the treatments (Supplementary Fig. 4b). Based on chromosomal distribution (Fig. 1b), eccDNAs are widely distributed across the rice genome, with the highest eccDNA density in pericentromeric and centromeric regions (Fig. 1c). Furthermore, these high-density regions also overlap with areas of high transposable elements (TEs) and DNA methylation density.

During eccDNA identification with ecc_finder in each sample, redundancy has already been optimized[27], and enabled the assignment of unique IDs for eccDNA regions. However, when generating IDs across all samples, certain genomic regions can lead to the artificial duplication of IDs. For further characterization, we designed a process to minimize ID duplication and ensure accurate ID assignment across samples. Briefly, we assigned the same ID to the eccDNA regions with overlaps of more than 200 bp among samples, preserving the full diversity of eccDNAs and ensuring that the original loci from ecc_finder's output remained unchanged (Supplementary Fig. 5a). From the initial 140,619 eccDNAs, we obtained 96,757 eccDNA IDs (Supplementary Fig. 5b). These ID-eccDNAs were classified into different genomic features, namely genes (5'UTR, CDS, 3'UTR), 2 kb upstream of the genes (up2kb), 2 kb downstream of the genes (down2kb), and intergenic regions. 30.3% of ID-eccDNAs overlapped gene regions (*ecGenes*) (Fig. 1d), of which 267 covered full-length genes. 15.8% of ID-eccDNAs mapped to up2kb regions, 8.8% to down2kb, and 45.1% to intergenic regions. A more detailed analysis of eccDNAs classified as genes showed that 40.1% overlapped to CDS regions (Fig. 1g), 18.9% to 3'UTRs, and 8.0% to 5'UTRs.

When mapping to transposable elements or non-coding RNAs, 51.2% ID-eccDNAs overlapped to TEs (*ecTEs*) (Fig. 1e), 40.3% to non-coding RNAs (*ecNon-codingRNAs*) (Fig. 1f). Most ID-eccDNAs mapped to single genomic features, however, 3,504 originated from regions encompassing genes, TEs, and non-coding RNAs (Fig. 1h) and 5,668 to regions covering up2kb, TEs, and non-coding RNAs (Fig. 1i). Additionally, we identified 2843 ID-eccDNAs covering a region spanning from the down2kb, TEs, and non-coding RNAs (Fig. 1j).

Although the organellar genome was removed from the reference during eccDNA identification, the regions of nuclear mitochondrial DNA (NUMT) and nuclear plastid DNA (NUPT) were kept in the chromosomal genome[28] for this analysis. We specifically determined the

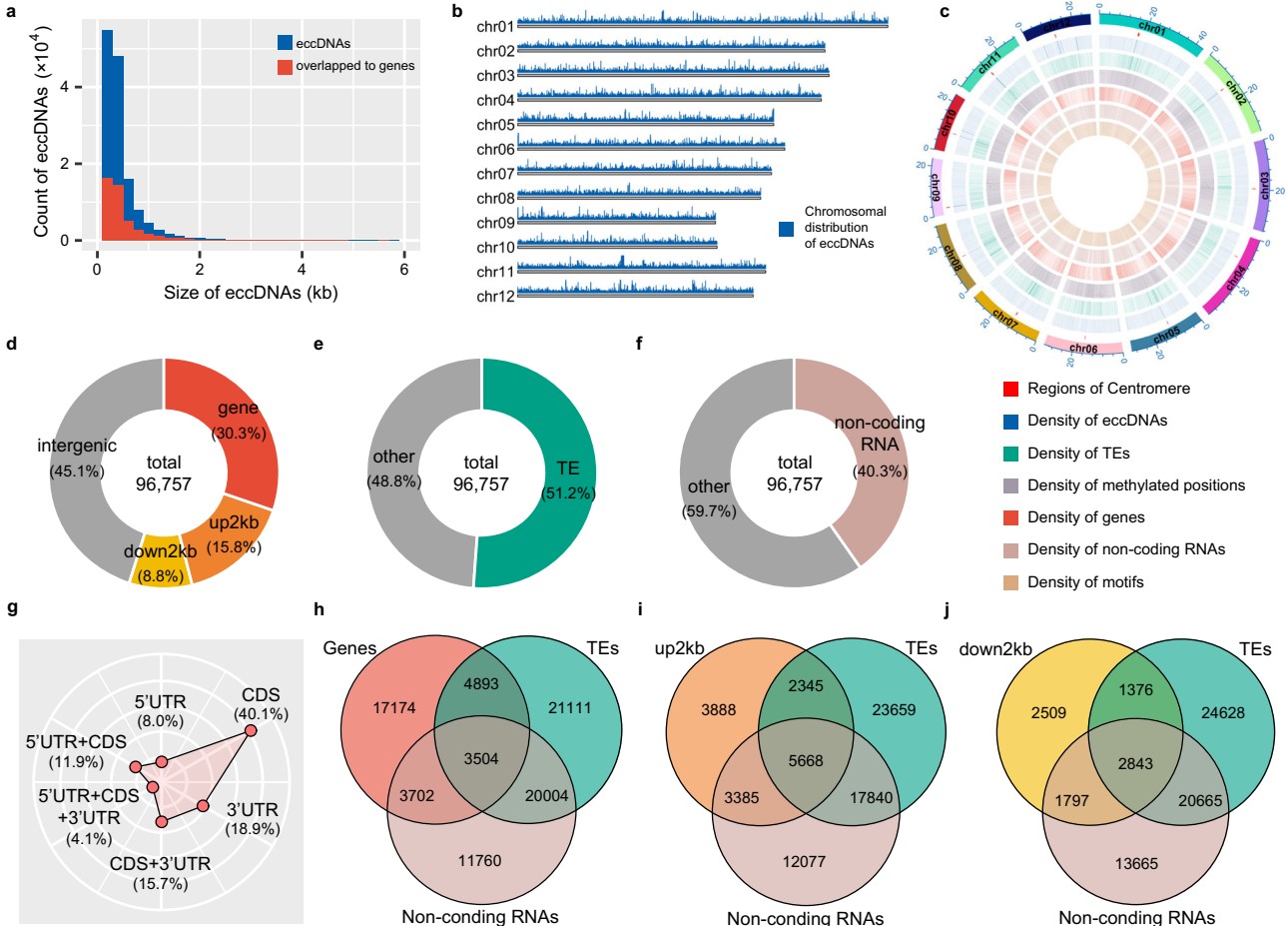

**Fig. 1 | Genome-wide distribution and global characterization of rice eccDNAs. a** Size distribution (kb) of eccDNAs (blue bars) and eccDNAs that overlap with genes (red bars) detected in all 16 samples. Only the most abundant eccDNAs ranging from 200 bp to 6 kb are shown. **b** Karyoplot displaying the chromosomal distribution of eccDNAs found in all samples. **c** Density distribution (in 100 kb windows) of eccDNAs in all samples, TEs, methylated positions, genes, non-coding RNAs, and motifs across the 12 chromosomes of rice. **d** Proportion of overall eccDNAs overlapping with genes (*ecGenes*), upstream 2 kb regions (up2kb), downstream 2 kb regions (down2kb), and intergenic regions. **e** Proportion of overall eccDNAs overlapping with Transposable Elements (*ecTEs*). **f** Proportion of

overall eccDNAs overlapping with non-coding RNAs (*ecNon-codingRNAs*). **g** Proportion of all *ecGenes* overlapping with unique 5′UTR, unique CDS, unique 3′UTR, 5′UTR together with CDS (5′UTR + CDS), CDS together with 3′UTR (CDS + 3′UTR), and all together with 5′UTR, CDS, and 3′UTR (5′UTR + CDS + 3′UTR) regions. **h** Venn diagram showing all *ecGenes* (Genes in red), *ecTEs* (TEs in green), and *ecNon-codingRNAs* (Non-coding RNAs in pink). **i** Venn diagram showing overall eccDNAs overlapping with upstream 2 kb regions (up2kb in orange), *ecTEs* (TEs in green), and *ecNon-codingRNAs* (Non-coding RNAs in pink). **j** Venn diagram showing overall eccDNAs overlapping with downstream 2 kb regions (down2kb in yellow), *ecTEs* (TEs in green), and *ecNon-codingRNAs* (non-coding RNAs in pink).

proportion of ID-eccDNAs that originated from NUMTs and NUPTs. Our analysis showed that only 0.82% of ID-eccDNAs were derived from NUMTs and 0.64% from NUPTs (Supplementary Fig. 5c), which agrees with previously published data[17]. Among ID-eccDNAs spanning NUMT and NUPT regions, we identified 17 classified as *ecGenes* (Supplementary Fig. 5d), 75 as *ecTEs* (Supplementary Fig. 5e), and 156 as *ecNon-codingRNAs* (Supplementary Fig. 5f). These data demonstrate that the rice eccDNA landscape is widely spread across the rice chromosomal genome, with a higher density in TE and centromeric regions.

### Gene-overlapped eccDNAs (*ecGenes*) show dynamic changes during rice growth

To explore the dynamic changes of eccDNA production during rice growth, we investigated *ecGenes* at three stages of rice development in optimal growth conditions: short-term (*Ctrl_D1* vs. *Ctrl_D3*), mid-term (*Ctrl_D1* vs. *Ctrl_D7*), and long-term (*Ctrl_D1* vs. *Ctrl_D14*). After performing GMPR normalization and EMDomics differential analysis, we identified *ecGenes* as differentially enriched if their *q*-value < 0.05 (see "Methods"). We further classified differentially enriched *ecGenes* as exclusive if they were present in both biological replicates of one

treatment but absent in the two replicates of the other, and as differential if the number of reads for one *ecGene* was higher in one treatment than the other. We further classified differential and exclusive *ecGenes* based on Gene Ontology (GO) enrichment analysis and summarized using REVIGO (see "Methods). When comparing GO results of differential + exclusive *ecGenes* with exclusive *ecGenes* alone, we found no substantial differences in the total counts or the GO enriched categories (Supplementary Figs. 6 and Supplementary Data 2), reflecting that exclusive *ecGenes* were the most statistically significant.

In the short-term optimal growth stage, we identified 863 exclusive *ecGenes*, including five that contain full-length genes (Supplementary Data 3). Among exclusive *ecGenes*, 230 were unique to *Ctrl_D1* and 633 to *Ctrl_D3* (Fig. 2a). To better understand their functional roles, we performed GO enrichment analysis on exclusive *ecGenes* from both samples to identify eccDNAs associated with positive and negative effects on specific biological processes. GO enrichment analysis on these 863 exclusive *ecGenes* (Fig. 2d) revealed that enriched categories included 'reproductive system development' (2.86-fold) and 'developmental process involved in reproduction' (2.83-fold), as well as categories associated with metabolic processes, among which 'N

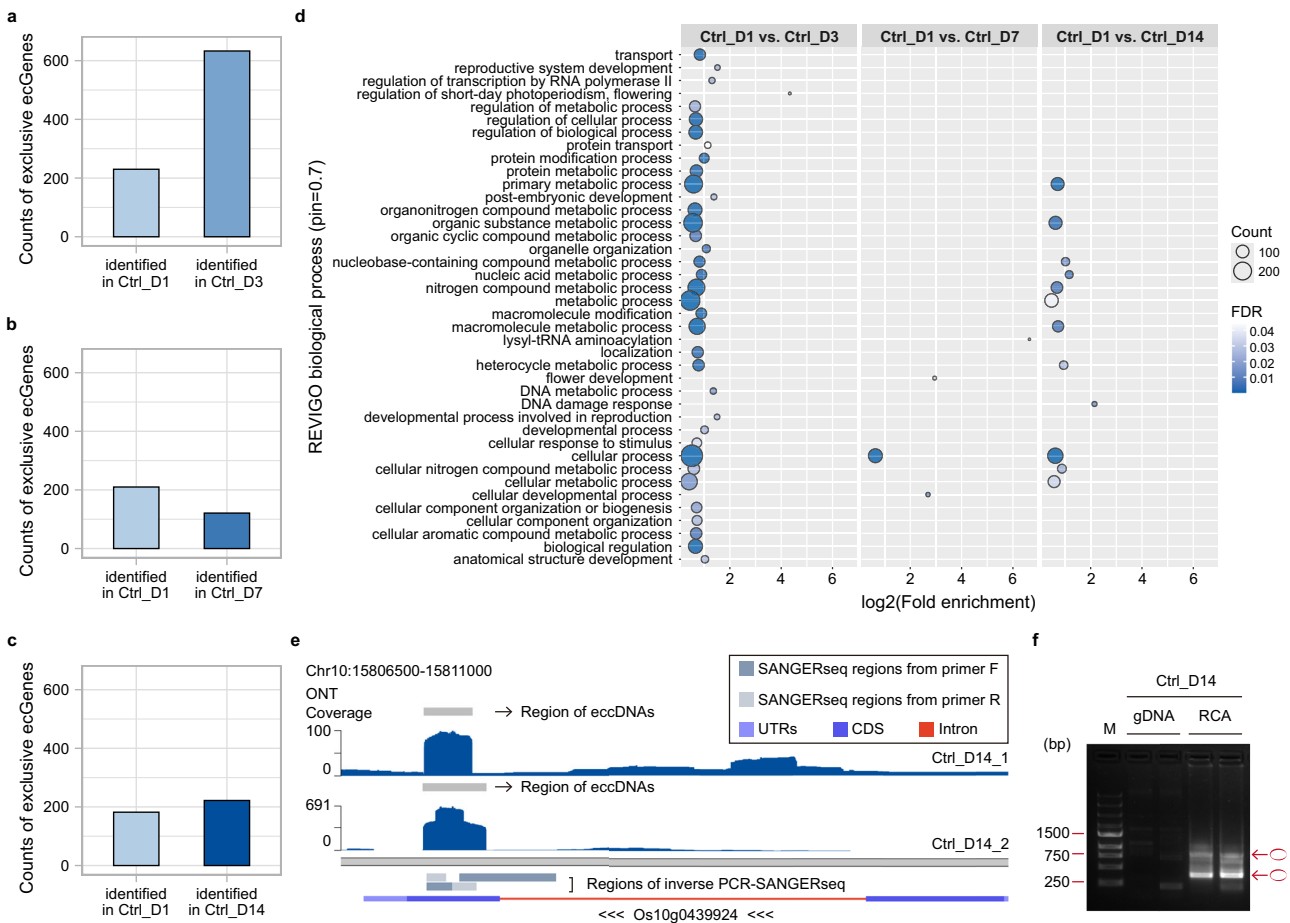

**Fig. 2 | Identification of exclusive *ecGenes* in rice during optimal growth.**
**a** Count distribution of identified exclusive *ecGenes* between 1-day control samples (*Ctrl_D1*) and 3-day control samples (*Ctrl_D3*). **b** Count distribution of identified exclusive *ecGenes* between 1-day control samples (*Ctrl_D1*) and 7-day control samples (*Ctrl_D7*). **c** Count distribution of identified exclusive *ecGenes* between 1-day control samples (*Ctrl_D1*) and 14-day control samples (*Ctrl_D14*). **d** Dot-plot illustrating the significantly enriched GO biological process categories summarized by REVIGO for the exclusive *ecGenes* during optimal growth from *Ctrl_D1 vs. Ctrl_D3*, *Ctrl_D1 vs. Ctrl_D7*, and *Ctrl_D1 vs. Ctrl_D14*. **e** Karyoplots of the Chr10:15806500-15811000 locus showing (from top to bottom) regions of eccDNAs and ONT

sequencing coverage in two replicates of 14-day control samples (*Ctrl_D14_1* and *Ctrl_D14_2*), regions of SANGER sequencing (SANGERseq) fragments from inverse PCR validation in 14-day control samples, and the transcript structure of Os10g0439924. **f** Gel electrophoresis of inverse PCR products from the *ecGene* derived from the Os10g0439924 locus in 14-day control samples (*Ctrl_D14*). The experiment was conducted twice. M: DNA Marker; bp: base pair; gDNA: template genomic DNA from *Ctrl_D14*; RCA: template rRCA products from *Ctrl_D14*; forward primer: 5′-CCCTACATATTGCCCATGCATTCAC-3′; reverse primer: 5′-GTATTGGAA-GAGTGCCGAGGAG-3′. Circles in red: symbol used for pointing out amplified products of *ecGene*.

compound metabolic process' was significantly represented with 177 enriched *ecGenes* (Supplementary Data 4). Notably, there is significant enrichment in the 'regulation of short-day photoperiodism/flowering' category, exhibiting the highest fold enrichment value of 20.19 (Supplementary Data 4). This category includes *ecGenes* mapping to *OsCO3*, *OsMADS51*, and *SDG718*, all essential in regulating short-day flowering in rice[29–31].

In the mid-term optimal growth phase, we identified 331 exclusive *ecGenes*, including three containing full-length genes (Supplementary Data 3). Among these, 210 were unique to *Ctrl_D1* and 121 to *Ctrl_D7* (Fig. 2b). GO enrichment analysis of these *ecGenes* revealed four enriched categories (Fig. 2d). The cellular process category was observed for both the short-term and mid-term growth phases. The 'lysyl-tRNA aminoacylation' category had the highest fold enrichment value (>100), which may indicate the role of eccDNAs in regulating the mechanisms of transferring activated amino acids to the 3′-OH group of lysine-accepting tRNA in rice (Supplementary Data 5). It is worth mentioning that unlike the developmental pathways enriched in the short-term growth phase, the 'flower development' category was uniquely identified in the mid-term phase, suggesting that *ecGenes*

dynamics respond to specific developmental processes along initial growth.

In the long-term optimal growth stage, we identified 404 exclusive *ecGenes*, including five containing full-length genes (Supplementary Data 3). Among these, 182 were unique to *Ctrl_D1* and 222 to *Ctrl_D14* (Fig. 2c). GO enrichment analysis (Fig. 2d) revealed enrichment of the 'N compound metabolic process' category, though with fewer *ecGenes* (81) than in the short-term growth stage (177). Notably, the 'DNA damage response pathway' category had the highest fold enrichment value (4.44), suggesting a potential association between eccDNA formation mechanisms and DNA damage during long-term optimal growth[32].

The number of exclusive *ecGenes* across different growth stages showed significant fluctuations, decreasing sharply from the short-term to the mid-term growth stage but increasing from the mid-term to the long-term stage (Fig. 2a–c). Compared to the short-term growth stage, the number of significantly enriched pathways was lower in the mid-term and long-term growth stages. However, specific pathways showed higher fold enrichment values in the later growth phases. Among the GO enrichment related to cellular component, the 'cellular

anatomical entity' category was identified in all growth stages with similar fold enrichment values (1.31 in short-term growth, 1.36 in both mid-term and long-term growth) (Supplementary Fig. 7b). Moreover, in the long-term growth stage, 'transferase complex' was observed as the category with the highest fold enrichment value (3.00). Thus, our GO results indicate that *ecGenes* are enriched specifically at different growth stages, suggesting a role of *ecGenes* in rice development.

We validated one *ecGene* detected in 14-day control samples through inverse PCR and SANGER sequencing (SANGERseq) (Fig. 2e, f). This *ecGene* originated from the Os10g0439924 locus region and covered most of the CDS (Fig. 2e). When comparing the typical raw Oxford Nanopore sequencing (ONT) reads to the reference sequence of Os10g0439924, two types of looping structures were characterized (Supplementary Figs. 8a–c). These structures agreed with the SANGERseq results of two sets of inverse PCR products shown in electrophoresis analysis (Fig. 2f). Additionally, we validated another *ecGene* detected in all control samples through inverse PCR and SANGERseq (Supplementary Figs. 8e–f). This *ecGene* originated from the last intron of the Os06g0651100 locus and overlapped with a set of repeat units named *ORSgTEMT01602424* (Supplementary Fig. 8d). Considering that the sequence of these repeat units is not unique across the entire chromosomal genome, the comparison between typical raw ONT reads, and the genomic sequence of Os06g0651100 was performed to validate the detailed looping structure of this *ecGene* (Supplementary Fig. 8g).

## Gene-overlapped eccDNAs respond to N stress

N deficiency is the most prevalent environmental stress affecting plants in natural ecosystems[33]. According to our previous research[34], N concentration in shoot tissue usually starts to decrease rapidly on the third day after LN treatment, and a stronger transcriptional response to N starvation is activated after seven days of treatment. To investigate the role of eccDNAs under N limitation, we conducted LN treatments on rice plants in a short-term treatment of 3 days (*Ctrl_D3* vs. *LN_D3*) and a long-term of 7 days (*Ctrl_D7* vs. *LN_D7*). Consistent with the analysis of developmental stages, no statistical differences in GO-enriched categories were found when differential *ecGenes* were added to exclusive *ecGenes* for LN treatments (Supplementary Figs. 9a–b; and Supplementary Data 6), which further suggested that de novo formation rather than an increase in the abundance of *ecGenes* differentiate growth stages and responses to stress treatment.

We identified 2,424 exclusive *ecGenes* during short-term LN treatment, including six containing full-length genes (Supplementary Data 7). Among these, 639 were unique to *Ctrl_D3*, while 1785 were specific to *LN_D3* (Fig. 3a). GO enrichment analysis revealed significant enrichment in the 'N compound metabolic process' category, with 443 *ecGenes* (Supplementary Data 8; Fig. 3c). Compared to the same category during short-term optimal growth, the count of enriched *ecGenes* increased from 177 in optimal growth to 443 in LN treatment. Furthermore, we highlight several of these 443 enriched *ecGenes* with known functions that are related to N stress (Supplementary Data 8). Interestingly, we also identified the 'N compound transport' category under short-term LN treatment (Supplementary Data 8). These GO enrichment results indicate that, in response to LN stress, *ecGenes* abundance shifted from a potential developmental role to a more specific N starvation response. In addition to pathways related to N stress, we also found an enriched category in the 'phosphate-containing compound' metabolic process related to phosphorus metabolism. Moreover, significantly enriched categories of 'response to stimulus' and 'response to stress' suggest a relationship between exclusive *ecGenes* and the general response to abiotic stress (Supplementary Data 8; and Fig. 3c).

In the long-term LN treatment, we identified 1033 exclusive *ecGenes*, 29 of which contain full-length genes (Supplementary Data 7). Among these exclusive *ecGenes*, 101 were specific to *Ctrl_D7*, and 932

for *LN_D7* (Fig. 3b). GO enrichment analysis of these exclusive *ecGenes* showed that 'N compound transport' category was still enriched (Fig. 3d; and Supplementary Data 9), suggesting the roles of *ecGenes* in responding to long-term N stress. Additionally, the 'chloroplast protein-transporting ATPase activity' category was enriched in molecular function with the highest fold enrichment value of 45.34 (Supplementary Fig. 10a). This indicates that *ecGenes* are involved in chloroplast-related molecular functions to respond to the long-term N stress.

Compared to the short-term LN treatment, the number of exclusive *ecGenes* in long-term samples has decreased. However, the proportion of exclusive *ecGenes* detected only in *LN_D7* (932/1033) is higher than in *LN_D3* (1785/2424), with only 104 *ecGene* in common (Supplementary Fig. 9c). When considering the exclusive *ecGenes* detected in *Ctrl_D3* or *Ctrl_D7*, only 8 *ecGenes* were common to both developmental stages (Supplementary Fig. 9d). Although fewer categories from biological process were enriched during the long-term LN treatment, the 'N compound transport' was still statistically significant, revealing that specific *ecGenes* are produced in response to LN along different growth phases.

We validated one *ecGene* that showed significant induction in the long-term LN treatment, which overlaps with *OsNPF2.4* (Os03g0687000) (Fig. 3e). The chromosomal expression level of *OsNPF2.4* in old leaves is induced by N starvation and plays an important role in nitrate acquisition and long-distance transport[35]. Inverse PCR and SANGERseq analyzes showed that this *ecGene* is clearly detected in the 7-day LN sample and had the same sequence as that originally obtained from our eccDNA sequencing (Fig. 3f). The detailed looping structure of this *ecGene* was obtained by comparing typical raw ONT reads with the Os03g0687000 genomic sequence (Supplementary Fig. 9e).

We performed classic RNA-seq on short-term and long-term LN treatments to identify the relationship between chromosomal differentially expressed genes (DEGs) and exclusive *ecGenes*. During the short-term LN treatment, 546 chromosomal DEGs were identified, of which 32 were common to exclusive *ecGenes* (Supplementary Fig. 11a). When these 32 genes were categorized into up- and down-regulated DEGs specific to exclusive *Ctrl_D3* or *LN_D3* *ecGenes*, we observed that 15 exclusive *ecGenes* in *LN_D3* exhibited down-regulated expression of their chromosomal genes (Supplementary Fig. 11b). During the long-term LN treatment, we identified 2,550 chromosomal DEGs, 88 of which were also classified as exclusive *ecGenes* (Supplementary Fig. 11c). Among these, the corresponding chromosomal genes for 42 *LN_D7*-exclusive *ecGenes* were down-regulated (Supplementary Fig. 11d). Of the *LN_D7* exclusive *ecGenes* that correlated with differential gene expression, two covered the full-length chromosomal gene (Supplementary Figs. 11g–h). One is Os11g0634200, which is up-regulated in response to N starvation (Supplementary Figs. 11e, i), and the other is Os12g0573600, which is down-regulated (Supplementary Figs. 11f, i). These results suggest a complex relationship between the presence of genes or partial genes in eccDNA and the expression levels of the corresponding chromosomal genes.

## Gene-overlapped eccDNAs respond to P stress

In addition to N deficiency, plants often experience P deficiency in most soils due to low P availability[36]. Given the lower demand of P compared to N at early tillering stages, rice takes longer to respond to LP treatment at both the global transcription and phenotypic levels[37]. Transcriptome analysis indicates that the initial significant response of key P starvation-related genes in rice is observed 7 days after treatment[37]. Consequently, the short-term LP treatment for eccDNA analysis was set at 7 days (*Ctrl_D7* vs. *LP_D7*), while the long-term LP treatment was established at 14 days (*Ctrl_D14* vs. *LP_D14*). No statistical differences in GO-enriched categories were found when differential

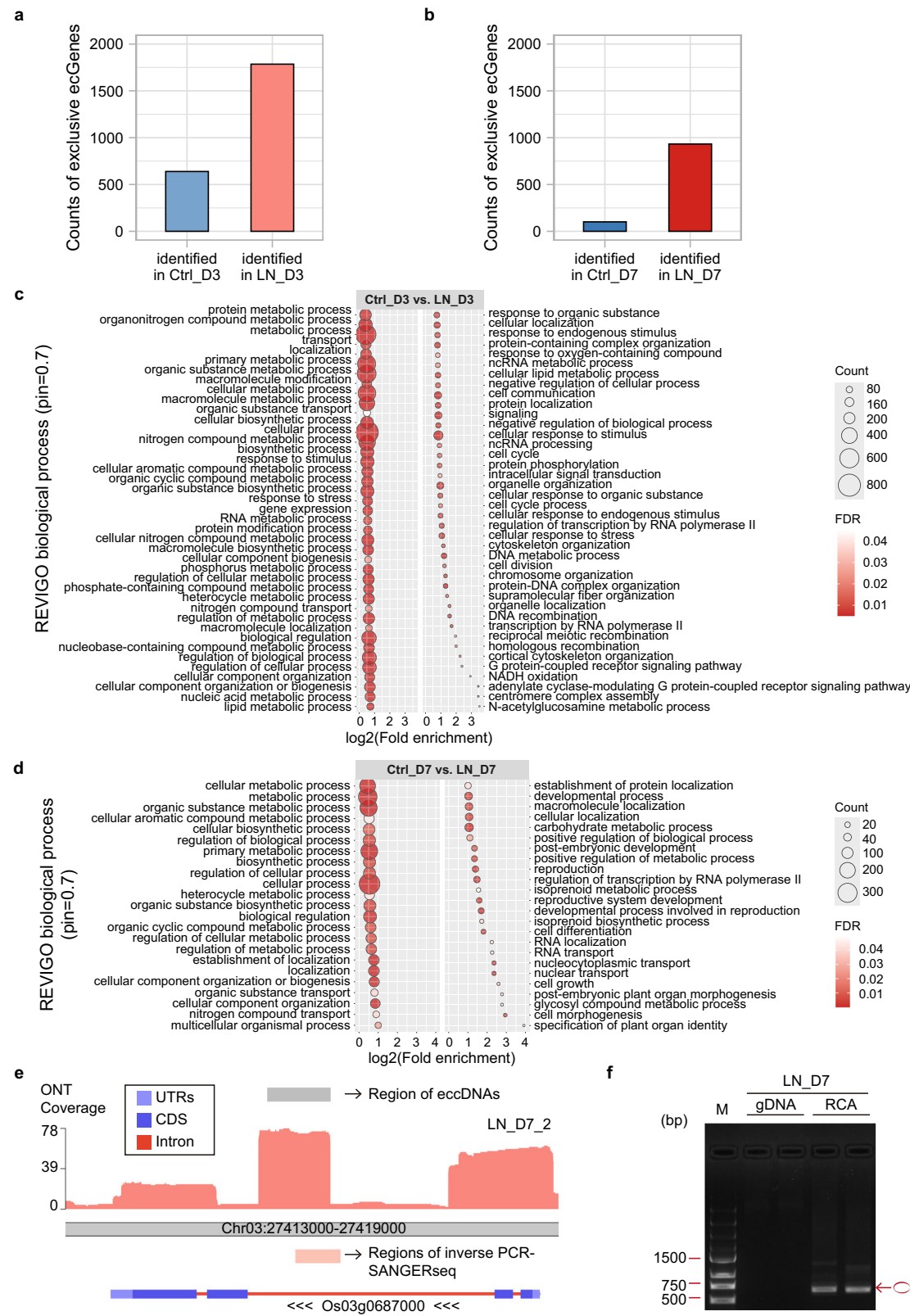

*ecGenes* were added to exclusive *ecGenes* for LP treatments (Supplementary Figs. 12a–b and Supplementary Data 10).

In the short-term LP treatment, we identified 556 exclusive *ecGenes*, two of which contain full-length genes (Supplementary Data 11). Among these, 91 were specific to *Ctrl_D7* and 465 to *LP_D7* (Fig. 4a). GO enrichment analysis (Fig. 4c) identified three highly enriched categories among those categorized as biological processes:

'tissue development' (fold enrichment value = 5.87), 'meristem maintenance' (fold enrichment value = 8.52), and 'meristem development' (fold enrichment value = 10.60). These categories highlight the potential roles of *ecGenes* in plant developmental responses to P deficiency, which is known to alter root and shoot architecture[36]. Additionally, we identified the 'N compound metabolic process' category, encompassing 100 *ecGenes* (Supplementary Data 12).

**Fig. 3 | Identification of exclusive *ecGenes* in response to low nitrogen (LN) in rice. a** Count distribution of identified exclusive *ecGenes* between 3-day control samples (*Ctrl_D3*) and 3-day LN treatment (*LN_D3*). **b** Count distribution of identified exclusive *ecGenes* between 7-day control samples (*Ctrl_D7*) and 7-day LN treatment (*LN_D7*). **c** Dot-plot illustrating the significantly enriched GO biological process categories summarized by REVIGO for the exclusive *ecGenes* from *Ctrl_D3* vs. *LN_D3*. **d** Dot-plot illustrating the enriched GO biological process categories summarized by REVIGO for the exclusive *ecGenes* from *Ctrl_D7* vs. *LN_D7*. **e** Karyoplots of the Chr03:27413000-27419000 locus showing (from top to bottom) region of

eccDNAs, ONT sequencing coverage in one replicate of 7-day LN treatment (*LN_D7_2*), regions of SANGER sequencing (SANGERseq) fragments from inverse PCR validation under 7-day LN treatments, and the transcript structure of Os03g0687000. **f** Gel electrophoresis of inverse PCR products from the *ecGene* derived from the Os03g0687000 locus under 7-day LN treatment (*LN_D7*). The experiment was conducted twice. M: DNA Marker; bp: base pair; gDNA: template genomic DNA from *LN_D7*; RCA: template rRCA products from *LN_D7*; forward primer: 5′-CAGAAGTGGCTCTCGCTATCAAAC-3′; reverse primer: 5′-GATAAT-CAGCCATAAGAGTGTTC-3′. Circle in red: amplified products of *ecGene*.

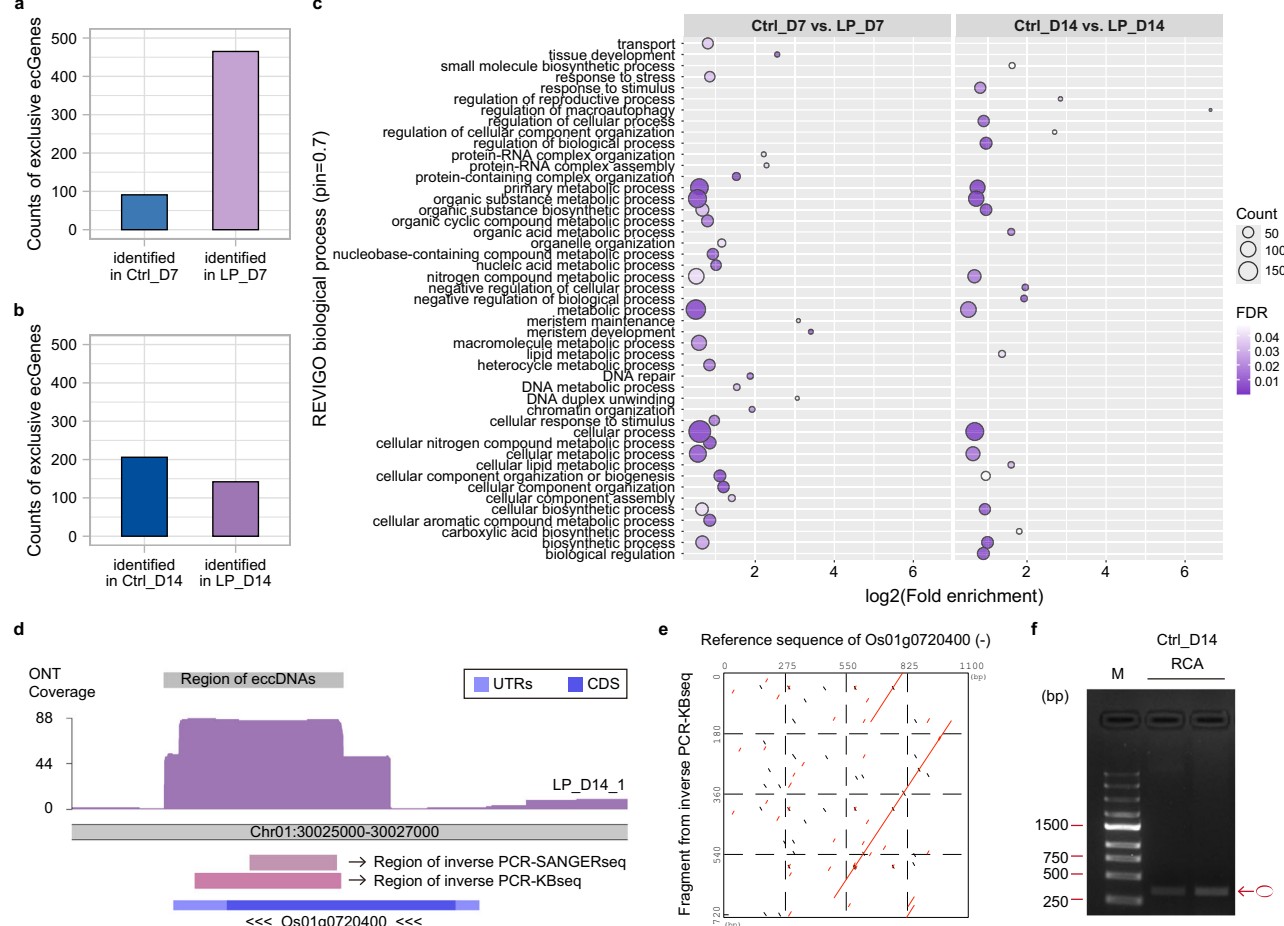

**Fig. 4 | Identification of exclusive *ecGenes* in response to low phosphorus (LP) in rice. a** Count distribution of identified exclusive *ecGenes* between 7-day control samples (*Ctrl_D7*) and 7-day LP treatment (*LP_D7*). **b** Count distribution of identified exclusive *ecGenes* between 14-day control samples (*Ctrl_D14*) and 14-day LP treatment (*LP_D14*). **c** Dot-plot illustrating the significantly enriched GO biological process categories summarized by REVIGO for the exclusive *ecGenes* from both *Ctrl_D7 vs. LP_D7* and *Ctrl_D14 vs. LP_D14*. **d** Karyoplots of the Chr01:30025000-30027000 locus showing (from top to bottom) region of eccDNAs, ONT coverage in one replicate of 14-day LP treatment (*LP_D14_1*), regions of SANGER sequencing (SAN-GERseq) fragments from inverse PCR validation under 14-day LP treatment, regions of KBseq (based on Illumina platform) fragments from inverse PCR validation under

14-day LP treatment, and transcript structure of Os01g0720400. **e** Dot-plot show-ing the correlation between individual sequences from KBseq of inverse PCR vali-dation under 14-day LP treatment and the reference sequence of Os01g0720400. Black dots: Seq2PlusStrand; Red dots: Seq2MinusStrand. **f** Gel electrophoresis of inverse PCR products from the *ecGene* derived from the Os01g0720400 locus under 14-day LP treatmen (*LP_D14*). The experiment was conducted twice. M: DNA Marker; bp: base pair; RCA: template rRCA products from *LP_D14*; forward primer: 5′-CAGGTTCCTCTCCTCGAGGAGCAC-3′; reverse primer: 5′-CTCGACTGCAAGCTC-GAGTCGTTG-3′. Circle in red: symbol used for pointing out amplified products of *ecGene*.

Interestingly, despite the absence of P-related categories in biological process, we highlight the 'monoatomic cation transmembrane trans-porter activity pathway' category, enriched in molecular function (Supplementary Fig. 13a). These findings suggest that eccDNAs may play a role in modulating membrane transporter activity in response to the short-term LP treatment (Supplementary Data 12).

In the long-term LP treatment, we identified 348 exclusive *ecGenes*, three of which contain full-length genes (Supplementary

Data 11). Among these exclusive *ecGenes*, 206 were detected only in *Ctrl_D14* and 142 in *LP_D14* (Fig. 4b). GO enrichment analysis (Fig. 4c) identified again the 'N compound metabolic process' category, encompassing 72 *ecGenes* (Supplementary Data 13). We identified the 'response to stimulus' category as the general response to abiotic stress. Notably, the highest fold enrichment was found for the 'reg-ulation of macroautophagy' category with a value over 100 (Supple-mentary Data 13). This category suggests a potential role for *ecGenes* in

modulating macroautophagy frequency, rate, or extent during long-term LP treatment. Within the enrichment of cellular component, we identified a high fold enrichment (43.97) for 'phosphatidylinositol 3-kinase complex' (Supplementary Fig. 13b). This indicates that during long-term LP treatment, *ecGenes* actively participate in modulating the expression of the phosphatidylinositol 3-kinase (PI3K) complex (Supplementary Data 13).

Comparing the long-term LP treatment to the short-term treatment, we found a decrease in the count and proportion of exclusive *ecGenes* in the long-term LP samples. Only five exclusive *ecGenes* were identified as common between *LP_D7* and *LP_D14* (Supplementary Fig. 12c). Notably, there were no common exclusive *ecGenes* between *Ctrl_D7* and *Ctrl_D14* (Supplementary Fig. 12d). The number of categories in GO enrichment of biological process and cellular component did not change significantly, and most categories related to metabolic process were enriched in both short-term and long-term LP treatment. However, specific pathways were still considerably enriched for each of the two low P treatments (Fig. 4c). The variation and abundance of *ecGenes* in response to LP treatment were generally less pronounced than in LN, regardless of the duration of P-starvation. However, a distinct and specific *ecGene* response to limited P supply was still evident in rice. Furthermore, based on the dynamics observed in GO enrichment results, we propose that extending the stress duration can modify the set of *ecGenes* involved in the response.

We performed inverse PCR, SANGERseq, and KBseq (based on Illumina platform) for one of the differential *ecGenes* identified in the long-term LP treatment spanning nearly the entire length of *OsACP1* (Os01g0720400) (Fig. 4d–e). This gene has been previously reported to play a role in rice P starvation[38]. Our results validated the presence of this *ecGene* in the 14-day LP sample (Fig. 4f). The dot plot between typical ONT reads and the reference sequence from *OsACP1* showed the detailed looping structures of this *ecGene* fragment (Supplementary Fig. 12e). Additionally, we found that the expression level of the chromosomal *OsACP1* was up-regulated at 14 days of LP treatments (Supplementary Fig. 14g).

Classic RNA-seq was performed on short-term and long-term LP treatments to clarify the association between chromosomal DEGs and exclusive *ecGenes*. During the short-term LP treatment, 10 of 858 chromosomal DEGs were also characterized as exclusive *ecGenes* (Supplementary Fig. 14a). Five were *LP_D7* exclusive *ecGenes* exhibiting down-regulated expression of their chromosomal genes (Supplementary Fig. 14b). In the long-term LP treatment, 1,088 chromosomal DEGs were characterized of which 14 were exclusive *ecGenes* (Supplementary Fig. 14c). Six exclusive *Ctrl_D14* *ecGenes* were down-regulated (Supplementary Fig. 14d). We analyzed one of the exclusive *ecGenes*, which covered the full-length gene of Os01g0373700 in at least one biological replicate of *Ctrl_D14* (Supplementary Fig. 14e–g). Together, the results in both LN and LP treatments suggest that *ecGenes* that overlapped to partial or full-length genes all display a complex association with chromosomal gene expression, while also being assumed to involve different regulatory mechanisms.

## Nutritional stresses lead to variation in TE-overlapped eccDNAs in rice

The relationship between eccDNAs and TEs has been recently studied in several organisms, particularly in plants, such as Arabidopsis and potato[14,22,23]. We quantitatively analyzed eccDNAs that overlap with TEs (*ecTEs*) and full-length repeat units (*full-length ecRepeatUnits*) across all treatments. We identified 35,485 *ecTEs* and 6866 *full-length ecRepeatUnits* from all 16 samples spanning eight treatments (Supplementary Fig. 15a). Among the *ecTEs*, 611 were common to all treatments (Supplementary Fig. 15b).

Over 70% of *ecTEs* were DNA transposons, with over 60% belonging to the *Zator* and *hAT* families (Fig. 5a, b). The proportion of DNA transposons in *ecTEs* did not change significantly during short-

term and mid-term optimal growth (Fig. 5b). However, under LN treatment, we highlight a substantial change in the ratio of DNA transposons to retrotransposons (Fig. 5b, d). The ratio of retrotransposons increased from 16.57% in *Ctrl_D3* and 16.79% in *Ctrl_D7* to 21.27% in the short-term LN treatment and 26.72% in the long-term LN treatment. This increase was more prevalent for the *Gypsy* family of LTRs (Fig. 5c, d). A similar trend was also observed for the LP treatments, in which the proportion of retrotransposons among *ecTEs* increased from 16.79% in *Ctrl_D7* to 17.47% in *LP_D7* and from 19.40% in *Ctrl_D14* to 20.94% in *LP_D14* (Fig. 5d). These findings suggest that both LN and LP treatments induce an increase in the ratio of retrotransposons in *ecTEs*, with more significant induction in longer treatments.

Among the *full-length ecRepeatUnits*, 68 were common to all treatments (Supplementary Fig. 15c). We then focused on the main DNA transposon families (Fig. 5e–g), especially *Stowaway*, which makes up about 2% of the rice genome[39,40]. We characterized 364 *full-length ecRepeatUnits* from *Stowaway* family. The count of *Stowaway-full-length ecRepeatUnits* was higher in all nutrient-stressed samples (226) than in all controls (138) (Supplementary Fig. 16a). However, prolonged nutritional stresses reduced the count of *Stowaway* units (from 134 in *LN_D3* to 19 in *LN_D7*, and from 47 in *LP_D7* to 4 in *LP_D14*). We identified 26 out of the 36 *Stowaway* subfamilies in the *full-length ecRepeatUnits* from all treatments (Supplementary Fig. 16a). Both LN and LP treatments influenced the number of subfamilies and the average reads per subfamily. Samples at 3 days had the highest count of subfamilies (20 in *Ctrl_D3*, 24 in *LN_D3*), followed by samples at 7 days (12 in *Ctrl_D7*, 13 in *LN_D7*, 16 in *LP_D7*), and 14 days (5 in *Ctrl_D14*, 3 in *LP_D14*), revealing a negative correlation between prolonged treatment or developmental stage with the number of *Stowaway* subfamilies (Supplementary Fig. 16a). It is also worth mentioning that *full-length ecRepeatUnits* of a specific set of *STOWAWAY11_OS*, located in the region on chromosome 7, were identified in all eight treatments (Supplementary Fig. 16b). Analysis of the chromosomal loci of repeat units and TEs revealed that an element from *Tc1_Mariner* superfamily overlaps with one of the *STOWAWAY11_OS* elements. A higher methylated position density was also detected in and around these *STOWAWAY11_OS* regions. Considering these findings along with previous studies on *Stowaway* and *Tc1_Mariner*[39,40], which suggest that Mariner-like transposases are required for *Stowaway's* transposable activity in rice, we propose a potential activity of *STOWAWAY11_OS full-length ecRepeatUnits*.

In addition to *Stowaway*, we identified another DNA transposon family, *Kiddo*, which was detected only in eccDNAs of LN and LP treatment samples at 3 and 7 days (Fig. 5e–g). We found 3 of the 4 *Kiddo* subfamilies in the *full-length ecRepeatUnits* from these treatments. The *ecRepeatUnits* in *LN_D3* samples covered a full-length *KIDDO_D* of chromosome 8 (Supplementary Fig. 15d), while in *LN_D7* samples, *ecRepeatUnits* covered two consecutive full-length *KIDDO_B* of chromosome 1 (Supplementary Fig. 15e). In addition, in *LP_D7* samples, a full-length *KIDDO_C* on chromosome 2 was covered by the eccDNAs together with another nearby repeat unit (Supplementary Fig. 15f). Previous studies suggested that *Kiddo* might still be active in the rice genome[41]. These results highlight the dynamic nature of *full-length ecRepeatUnits* linked to DNA transposon families in response to nutritional stresses and treatment duration, particularly emphasizing the prevalence and potential activity of the *Stowaway* and *Kiddo* elements in eccDNAs.

## Identification and validation of multiple-fragment eccDNAs in rice

Single fragment is typically the predominant eccDNA class found in all organisms examined to date[11] and in this study (Fig. 6a). However, a fraction of eccDNAs can have sequences derived from different genomic regions, consisting of either two or more non-contiguous

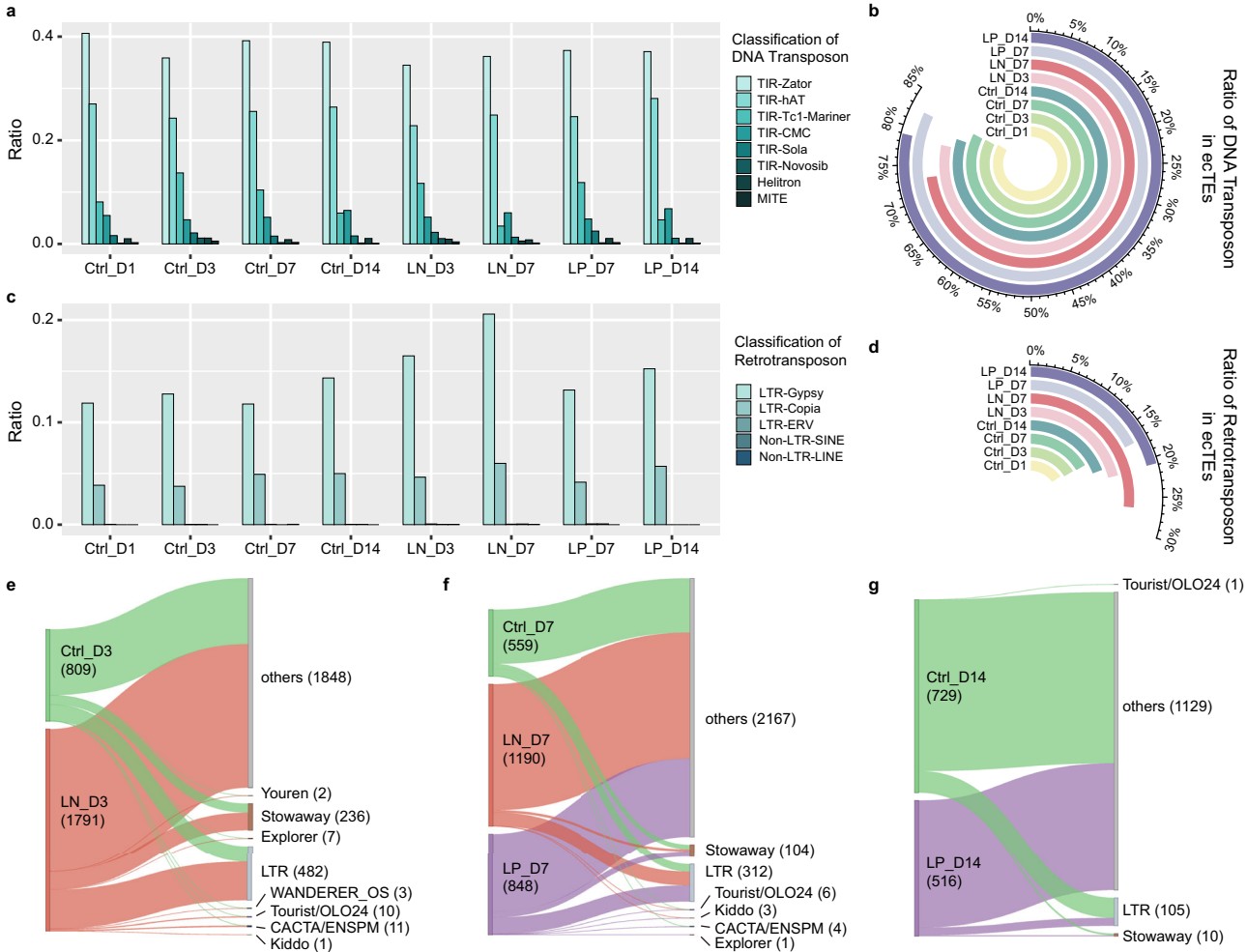

**Fig. 5 | Classification and characterization of *ecTEs* and *full-length ecRepeatUnits* in rice. a** Ratio distribution on DNA transposon classification in *ecTEs* across all treatments. **b** Proportion of DNA transposon in *ecTEs* for all treatments. **c** Ratio distribution on Retrotransposon classification in *ecTEs* across all treatments. **d** Proportion of Retrotransposon in *ecTEs* for all treatments. **e** Sankey diagram illustrating the distribution of full-length *ecRepeatUnits* in 3-day control samples (*Ctrl_D3*) *vs.* 3-day LN treatment (*LN_D3*) with connections to specific repeat unit families. **f** Sankey diagram illustrating the distribution of *full-length ecRepeatUnits* in 7-day control samples (*Ctrl_D7*) *vs.* 7-day LN treatment (*LN_D7*) and 7-day LP treatment (*LP_D7*) with connections to specific repeat unit families. **g** Sankey diagram illustrating the distribution of *full-length ecRepeatUnits* in 14-day control treatments (*Ctrl_D14*) *vs.* 14-day LP treatment (*LP_D14*) linking with specific repeat unit families.

fragments from the same chromosome, or two or more sequences from different chromosomes. Here, we refer to this kind of eccDNAs as multiple-fragment eccDNAs (MF-eccDNAs), as previously defined[11]. To complement the comprehensive understanding of rice eccDNAs, eccDNA_RCA_Nanopore[42], a tool reported with good performance in MF-eccDNA detection in human cells[11], was used for the identification of MF-eccDNAs in reads that mapped to the rice genome (Supplementary Figs. 3a, c). eccDNA_RCA_Nanopore contains a specific framework designed to detect cross-chromosomal or far-distance junctional reads from long-read sequencing[42]. When comparing *ecGenes* identified by eccDNA_RCA_Nanopore to ecc_finder, an average of 95.5% were detected by both pipelines among all samples, validating the reliability of the identified eccDNAs. Based on eccDNA_RCA_Nanopore analysis, 99.7% of eccDNA reads consisted of only one fragment (Nfragment = 1) (Fig. 6a), while the remaining 0.3% consisted of two to ten non-contiguous DNA regions (Nfragment ≥ 2).

We mapped all MF-eccDNA reads to the rice genome and grouped the results by core gene IDs and the number of fragments (Nfragment) to reduce redundancy[43] (see "Methods"). The origin of sequences present in MF-eccDNAs was widespread in the rice genome (Fig. 6b). A notable example is a set of MF-eccDNAs that range

from 3 to 5 fragments containing a common core gene ID, Os04g0343050 (cytochrome c oxidase subunit 2). This set of MF-eccDNAs had a high load in *LN_D7* (Fig. 6c, d). One unique looping structure of this 5-fragment eccDNA (5 f eccDNA) was characterized by comparing raw ONT reads with the reference sequence in the corresponding rice genomic regions (Fig. 6f, g). This structure was then validated by inverse PCR (Fig. 6h, i) and whole plasmid sequencing using ONT for both 7-day LN and 14-day LP treatments (Fig. 6j, k).

We also identified another set of MF-eccDNAs with a putative cytochrome P450 as the core gene (Os05g0372300), composed of 2 to 7 fragments, with the 5-fragment variant being the most common (Fig. 6e). These MF-eccDNAs were more prevalent in both *LN_D7* and *LP_D14* (Fig. 6c). It is also important to note that we did not identify MF-eccDNAs with Os04g0343050 in any samples of *Ctrl_D3* or *Ctrl_D7*. Similarly, MF-eccDNAs with Os05g0372300 were absent in our *Ctrl_D1* samples. Interestingly, when analyzing the chromosomal counterpart positions of fragments in these two sets of MF-eccDNAs, an LTR unit was observed on one side and a DNA transposon unit on the other (Supplementary Data 14), suggesting a mechanism for their origin based on the function of TEs in rice.

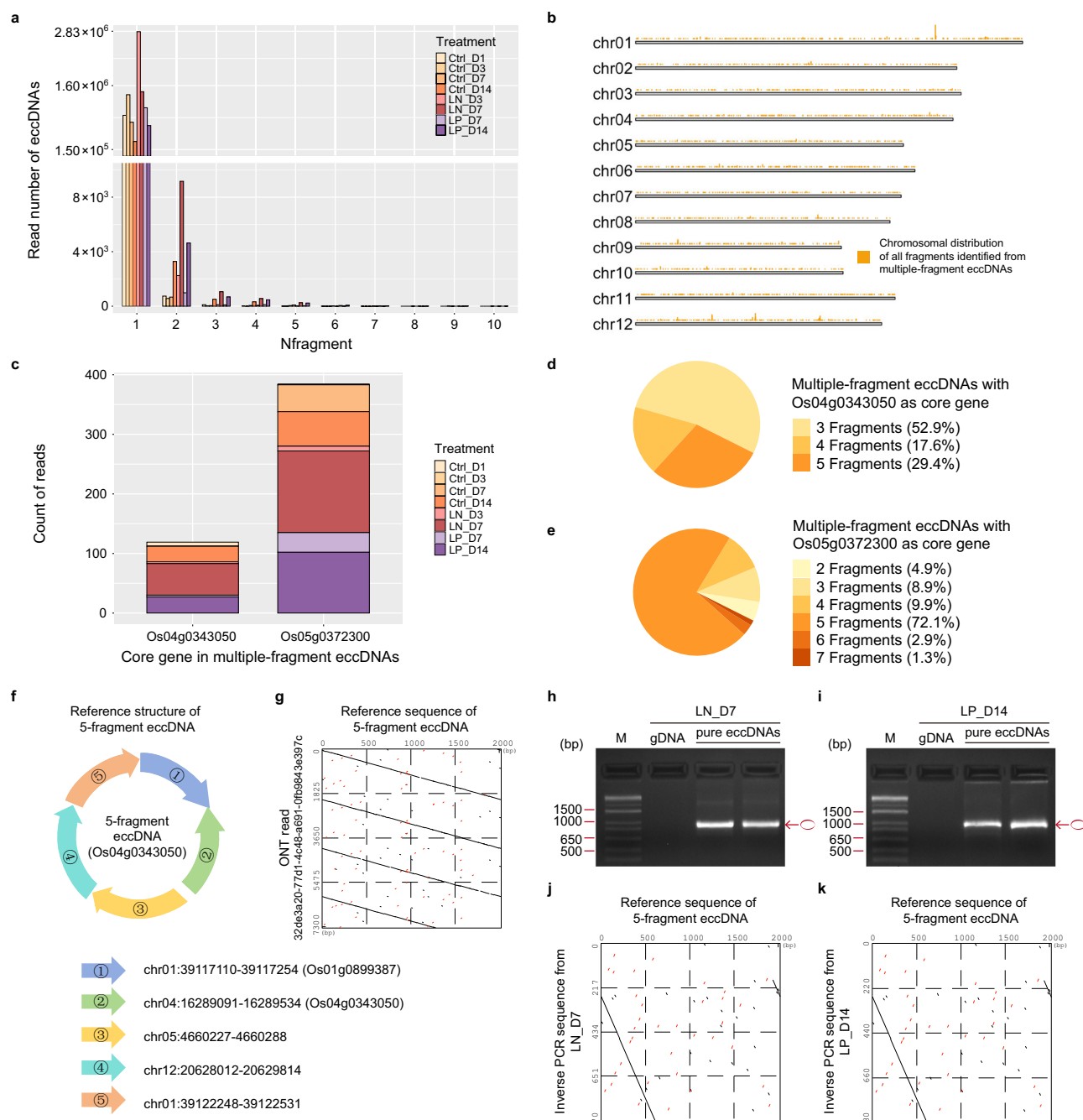

**Fig. 6 | Characterization of multiple-fragment eccDNAs (MF-eccDNAs). a** Count distribution of sequencing reads on the number of fragments (Nfragment) for eccDNAs by treatment. **b** Karyoplot showing the chromosomal distribution of all the fragments identified in the MF-eccDNA reads. **c** Count distribution on the reads of MF-eccDNAs with Os04g0343050 (cytochrome c oxidase subunit 2) or Os05g0372300 (putative cytochrome P450) as core gene by treatments. **d** Count distribution on the number of fragments in MF-eccDNAs with Os04g0343050 as the core gene identified in all treatments. **e** Count distribution on the number of fragments in MF-eccDNAs with Os05g0372300 as the core gene identified in all treatments. **f** Circular representation and detailed fragment location of the 5-fragment (5 f) eccDNAs with Os04g0343050 as the core gene. **g** Dot-plot showing the correlation between individual sequence from ONT sequencing read and reference sequence of Os04g0343050-5f eccDNAs under 14-day LP treatments. Black dots: Seq2PlusStrand; Red dots: Seq2MinusStrand; bp: base pair. **h** Gel electrophoresis of inverse PCR products from the Os04g0343050-5f eccDNAs under 7-day LN treatments (*LN_D7*). The experiment was conducted twice. M: DNA Marker; bp: base pair; gDNA: template genomic DNA from *LN_D7*; pure eccDNAs:

template pure eccDNAs from *LN_D7*; forward primer, 5′-GAAGACCA-CACTCAGGATGT-3′; reverse primer, 5′-CACTTAAATCACCCTCGCTAAGAG-3′. Circle in red: amplified products of eccDNA). **i** Gel electrophoresis of inverse PCR products from the Os04g0343050-5f eccDNAs under 14-day LP treatments (*LP_D14*). The experiment was conducted twice. M: DNA Marker; bp: base pair; gDNA: template genomic DNA from *LP_D14*; pure eccDNAs: template pure eccDNAs from *LP_D14*; forward primer, 5′-GAAGACCACACTCAGGATGT-3′; reverse primer, 5′-CACTTAAATCACCCTCGCTAAGAG-3′. Circle in red: amplified products of eccDNA. **j** Dot-plot showing the correlation between the individual sequence from Whole Plasmid Sequencing (based on ONT) of inverse PCR validation under 7-day LN treatments (*LN_D7*) and the reference sequence of Os04g0343050-5f eccDNAs. Black dots: Seq2PlusStrand; Red dots: Seq2MinusStrand. **k** Dot plot showing the correlation between individual sequence from Whole Plasmid Sequencing of inverse PCR validation under 14-day LP treatments (*LP_D14*) and reference sequence of Os04g0343050- 5 f eccDNAs. Black dots: Seq2PlusStrand; Red dots: Seq2MinusStrand.

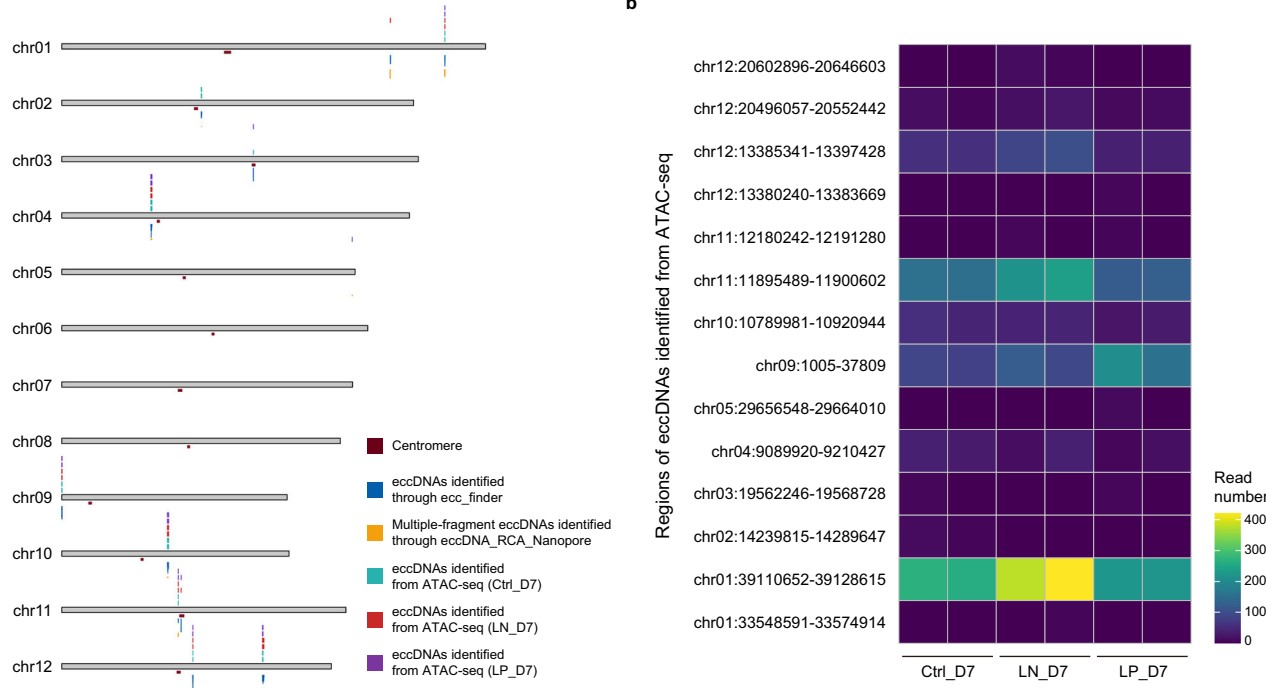

**Fig. 7 | ATAC-seq validation of high-density eccDNA regions in rice. a** Karyoplot showing the chromosomal distribution of high-density eccDNA regions validated through the ATAC-seq-ecc_finder pipeline, along with the distribution of fragments generated from single-fragment eccDNAs (ONT sequencing-ecc_finder pipeline) and multiple-fragment eccDNAs (ONT sequencing-eccDNA_RCA_ONT pipeline) in the same regions. **b** Heat map displaying the read count distribution from representative regions of eccDNAs detected by ATAC-seq-ecc_finder pipeline in 7-day control samples (*Ctrl_D7*), 7-day LN treatments (*LN_D7*) and 7-day LP treatments (*LP_D7*).

On the other hand, two sets of MF-eccDNAs with Os01g0791033 or Os12g0423313 as core genes appeared in all samples, predominantly composed of 2-fragment types (Supplementary Figs. 17a–e). As shown in the karyoplots (Supplementary Fig. 17f–g), some pairs of fragments were positioned adjacent to and flanked by multiple LTR units from the same subfamilies (Supplementary Data 14), suggesting a potential LTR-related homologous recombination (HR) mechanism in MF-eccDNA formation[13] (Supplementary Fig. 18).

An increased count of these MF-eccDNAs during longer growth (Supplementary Figs. 17a and Fig. 6c) suggests that an extended optimal growth phase leads to the accumulation of specific MF-eccDNAs. We also observed an increase in the number of reads of these MF-eccDNA sets in both *LN_D3* and *LN_D7* (Supplementary Figs. 17a and Fig. 6c), being much higher in *LN_D7*. Although not as pronounced as under LN stress, we observed that MF-eccDNA repertoire and abundance are influenced by P status in rice. These findings indicate that nutrient stress induces the accumulation of MF-eccDNAs, underscoring the dynamic genomic responses of rice to nutritional signals.

### ATAC-seq effectively validates high-density regions of rice eccDNAs

To further validate the high eccDNA density regions in rice, we performed the Assay for Transposase-Accessible Chromatin (ATAC-seq) using high-throughput sequencing on *Ctrl_D7*, *LN_D7* and *LP_D7*. As reported in other organisms, circular DNA molecules are retained during ATAC-seq library preparation and can be identified through split-read analysis in sequencing data[44,45]. However, because ATAC-seq does not specifically enrich circular DNA, it is limited to detecting high-density regions of eccDNAs[45]. While ATAC-seq has been used for eccDNA detection in animal cells, its application in plants remains unexplored.

Considering the requirement of a more precise framework for eccDNA identification in the plant genome, the short-read mode[43,45]

from ecc_finder was applied to ATAC-seq clean data (Supplementary Data 15) with the rice chromosomal genome as a reference. As shown in the analysis (Fig. 7a), 14 regions were characterized as high-density regions of single-fragment eccDNAs (ecc_finder) or MF-eccDNAs (eccDNA_RCA_Nanopore), and five of these mapped inside or near centromeres, which our research has identified as typical high-density eccDNA regions in rice (Figs. 6i and 1c). Seven regions did not exhibit significant differences between treatments (Fig. 6j). However, the read number of eccDNAs, such as those overlapping with chr01:39110652-39128615 or chr11:11895489-11900602, increased substantially in *LN_D7* when compared to *Ctrl_D7* or *LP_D7* (Fig. 7b).

Since Kumar et al.[44] and Kang et al.[45] reported the potential of ATAC-seq to identify eccDNAs, this technique was previously validated in human cancer cell lines, and now in rice in our work. Thus, this approach provides valuable insights into eccDNA dynamics in plants and offers strategies and tools for future research.

### Discussion

Here, we comprehensively analyze single-fragment and multiple-fragment eccDNAs (MF-eccDNAs) in rice. We characterized the presence and abundance of eccDNAs throughout plant development under optimal growth conditions and their dynamics under both short-term and long-term N and P stresses. Our findings showed that rice eccDNAs could contribute to plant development and responses to nutritional stresses by altering the abundance of *ecGenes* and *ecTEs*.

Recently, extensive research to characterize eccDNAs in various organisms has been conducted[6,11,15,17]. Our ONT sequencing data uncover the eccDNA size distribution in rice, with the majority ranging between 200 bp and 600 bp (Fig. 1a). Our results are consistent with a previous study on rice eccDNAs using short Illumina reads[17]. However, we identified a 4-fold higher number of distinct eccDNAs in rice shoots (96,757) than that previously reported (22,183)[17], likely due to the analysis strategy of samples subjected to nutritional treatment and the

long read sequencing platform used. Additionally, we observed a high percentage of eccDNAs mapping to CDS regions (Fig. 1g), which differs from findings in human cancer cell lines, where most eccDNAs originate from 5′UTR regions[3].

## LTRs in eccDNA dynamics suggest a hypothetical mechanism of origin

Previous reports have shown a strong relationship between eccDNAs and TEs in rice[14] and Arabidopsis[23]. LTRs have garnered particular attention due to their higher activity potential compared to other TEs and their ability to circularize through DNA repair mechanisms[14,22]. Our study indicates that nutritional stresses increase the abundance of LTRs in *ecTEs* (Fig. 5c, d). Although we did not directly evaluate the activity of these LTRs, our findings suggest the involvement of active retrotransposons in the variation of eccDNAs under nutritional stresses.

Our analysis of MF-eccDNAs suggests a mechanism of HR based on retrotransposons or remnants of LTRs (Supplementary Fig. 18). According to the karyoplots (Supplementary Fig. 17f), two regions of a unique set of MF-eccDNAs are flanked by several LTRs from the same family, *Os4_05_6L*. Moreover, the gap region between these two fragments skipped a long LTR from *Os4_05_6L*, suggesting a mechanism for generating some rice eccDNAs, similar to the mechanism previously reported for the formation of the *GAP^circle* in yeast[13]. Based on the formation mechanism of single-fragment eccDNAs in yeast, our findings support the hypothesis of LTR-mediated HR mechanism producing MF-eccDNAs in rice and probably other plant species (Supplementary Fig. 18).

## Potential roles of *Stowaway* and *Kiddo* DNA transposons in eccDNA dynamics

Unlike previous studies on eccDNA dynamics in rice[17], our findings highlight the significant accumulation of DNA transposons rather than retrotransposons. We observed the widespread identification of the *Stowaway* family in *full-length ecRepeatUnits* (Figs. 5e–g and Supplementary Fig. 16a). Although our classification in *ecTEs* did not reveal a significant number of miniature inverted-repeat transposable elements (MITEs), which include the *Stowaway* family[46], we noted an enrichment of *Tc1-Mariner* elements across all treatments (Fig. 5a). Previous studies have shown that *Tc1-Mariner* elements share similar terminal inverted repeats (TIR) and target site duplications (TSD) with *Stowaway*[40]. A complex relationship between *Tc1-Mariner* and *Stowaway* elements has been previously reported[39], which could clarify the discrepancy in classification. Moreover, the identification of *STOWAWAY11_OS* elements in *full-length ecRepeatUnits* from all treatments showed the close association between *Stowaway* and *Tc1-Mariner* elements (Supplementary Fig. 16b). Although the mechanisms behind *Stowaway*'s activity in plants are not yet fully understood, it has been shown that *Stowaway* elements interact with Mariner-like transposases to enable transposable activity in rice[40]. Given the accumulation of *Tc1-Mariner* in *ecTEs* and *Stowaway* elements in the *full-length ecRepeatUnits*, we propose that these *Stowaway* units, such as *STOWAWAY11_OS*, capable of forming circular molecules, may be active in rice.

The regulatory function of the MITE-type DNA transposon *Kiddo*, located in the rice ubiquitin2 promoter, was previously reported[41]. This work demonstrated the role of *Kiddo* elements in regulating gene expression in vitro and showed that sequence-specific epigenetic modifications reduce the transcriptional enhancement of *Kiddo* in vivo[47]. A comparison of different rice cultivars revealed that in some cultivars, *Kiddo* had been excised from the ubiquitin2 promoter (or inserted in some cultivars), indicating that at least some members of the MITE family are active and capable of excising from genomic DNA[41]. Our *full-length ecRepeatUnits* analysis identified *Kiddo* in nutritionally stressed samples (Supplementary Figs. 15d–f), indicating that

*Kiddo* can form circular molecules in response to environmental signals like nutrient availability (Fig. 5e–g). This further supports the potential activity of *Kiddo-full-length ecRepeatUnits* in rice. While further research is needed to determine how these circular units function under nutritional stress and whether they contribute to local or distant changes in gene expression, the observed increase in TE-related eccDNAs in the *ddm1* Arabidopsis mutant supports the hypothesis that DNA methylation plays a role in the formation of MITE-family *full-length ecRepeatUnits*[23].

## eccDNAs overlapped to functional genes are involved in rice growth and responses to N and P stresses

Our research highlights the potential role of gene-overlapped eccDNAs (*ecGenes*) in rice development and their response to nutritional stress. Generally, most exclusive *ecGenes* contain gene segments, with only a few containing full-length genes (Supplementary Data 3, 7, 11). Future studies are needed to determine the detailed biological significance of these exclusive *ecGenes* and the mechanisms by which *ecGenes* may modulate gene expression or induce epigenetic changes.

A GO analysis of exclusive *ecGenes* revealed that the enrichment of biological process categories varies across growth stages (Fig. 2d). During mid-term growth, we observed significant categories related to lysyl-tRNA aminoacylation (biological process) and lysine-tRNA ligase activity (molecular function), both showing exceptionally high fold enrichment values (>100) (Fig. 2d and Supplementary Fig. 7a). Lysyl-tRNA synthetases (LysRS) are thought to be nonhomologous duplications of aminoacyl-tRNA synthetases and, in some instances, assist in the aminoacylation of the rare amino acid pyrrolysine[48,49]. During the long-term growth phase, we observed that the 'DNA damage response pathway' exhibited the highest fold enrichment value (4.44), suggesting that eccDNAs may help prevent DNA damage during the reproductive stage or act as a response to environmental factors causing DNA damage during normal development. Our findings also indicate that exclusive *ecGenes* are more abundant during the early stages of growth. Further research is needed to clarify the biological function and the mechanisms by which these *ecGenes* contribute to rice growth and development.

When subjected to LN stress, changes in *ecGenes* were more pronounced than those observed during optimal growth. In short-term LN treatment, 'N compound metabolic process' showed a much higher accumulation of exclusive *ecGenes* (443) (Fig. 3c), several of which gene IDs were associated with LN stress (Supplementary Data 8). For instance, we identified *ecGenes* containing segments of *OsTIR1*, whose chromosomal gene expression is regulated by the accumulation of OsmiR393. Nitrogen-induced expression of *OsTIR1* promotes rice tillering[50]. Additionally, we discovered *ecGenes* derived from *OsGO-GAT1*, a gene encoding a central enzyme in ammonium assimilation in rice, which works in conjunction with *OsAMT1;2* to enhance rice survival under LN stress[51]. We also detected *ecGenes* associated with *OsNAR2.1*, a component of the high-affinity nitrate transporter system *OsNRT2.1/2.2* and *OsNRT2.3*[52]. *OsNAR2.1* facilitates nitrate absorption across various concentration levels and regulates auxin transport from shoot to root as part of the rice nitrate signaling system[53,54]. Moreover, we identified *ecGenes* overlapping with *OsNIT2*, a gene activated by *OsNAR2.1* that may contribute to root growth maintenance under varied N forms and concentrations[55]. These findings suggest that *ecGenes* play critical roles in the rice response to LN stress, highlighting their importance in N metabolism and signaling pathways essential for plant adaptation and survival under short-term N-limited conditions.

In addition to the 'N compound metabolic process' category, we identified several enriched *ecGenes* in the 'N compound transport pathway' during both short-term (Supplementary Data 8) and long-term (Supplementary Data 9) LN treatments. In short-term LN treatment, exclusive *ecGenes* enriched in 'N compound transport' included Os06g0633800, a putative amino acid transporter. We also noted

enriched *ecGenes* of the peptide transporter gene *OsPTR3* (*OsNPF5.5*) in this category[56]. During the long-term treatment, the profile of exclusive *ecGenes* enriched in N compound transport shifted. We identified exclusive *ecGenes* that overlap with another putative peptide transporter gene, *PTR2* (Os05g0431700), and an amino acid transporter gene, *OsGAT3*, known for its role in gamma-aminobutyric acid transport[57]. Also, we identified two amino acid permease genes among the exclusive *ecGenes*, *OsAAP11* (Os11g0195600) and *OsAAP3*, which have been shown to significantly improve N use efficiency[58]. These findings underscore the specific response of rice *ecGenes* to environmental LN pressure, highlighting the dynamic regulation of N compound transport and the potentially crucial role of *ecGenes* in adaptation to N-limited conditions.

In both short-term (7 days) and long-term (14 days) LP treatments, we observed enrichment of the 'N compound metabolic process' category. However, a comparison of the enriched exclusive *ecGenes* in this category of LP to those of LN (Supplementary Data 8, 12, and 13) revealed distinct gene IDs, indicating differences in the *ecGenes* response to different nutritional stress. During the short-term LP treatment, GO enrichment analysis highlighted *OsPI4K2* in the catalytic activity category. OsPI4K2, or β-type Phosphatidylinositol 4-kinase, is critical for membrane targeting and phospholipid binding, which is part of the lipid remodeling process during P starvation response[59]. In the long-term treatment, we identified the PI3K complex category in cellular component (Supplementary Data 13). Although the functions of the genes enriched in this category have not been extensively studied, PI3K plays an important role in growth regulation and stress responses, acting as a positive regulator of gibberellin (GA) signaling and abscisic acid (ABA)-induced hydrogen peroxide production in rice leaves[60–62]. We also found exclusive *ecGenes* overlapping with *OsPHT4;3*, a member of the OsPHT4 family of phosphate transporters[63]. Thus, our study reveals that *ecGenes* could play specific functions in the LP rescue response of rice.

## Potential functions of ecGenes in regulating chromosomal gene expression

Although a previous study suggested a minimal impact of rice eccDNAs on gene expression[17], our analysis revealed the complex association between partial or full-length exclusive *ecGenes* and the expression of their chromosomal counterparts in response to nutritional stresses (Supplementary Figs. 11 and 14). A key finding of our work is that half of the common gene IDs associated with *ecGenes* exclusive to LN or short-term LP treatments showed increased presence and were also identified as down-regulated DEGs (15/32 in short-term LN treatment; 42/88 in long-term LN treatment; 5/10 in short-term LP treatment). These results suggest that *ecGenes* may play a role in negatively regulating the expression of chromosomal genes. Studies on small eccDNAs (200–600 bp) in mammalian cell lines[10] have shown that eccDNAs containing exon fragments can repress host gene expression by producing microRNAs or siRNAs. Based on these findings, we propose that a similar mechanism of negative regulation by *ecGenes* in rice may involve microRNAs or siRNAs. However, consensus-based and executable protocols in rice have not been found recently to determine whether siRNAs or microRNAs are produced from eccDNA molecules. Future research should focus on developing effective strategies to identify the prospective transcripts from eccDNA molecules and also validate their functions on local or distant gene regulations.

We further observed other associations between exclusive *ecGenes* and chromosomal DEGs, beyond the repressing tendency (Supplementary Figs. 11e–i and 14e–f). We found one *ecGene* covering full-length Os11g0634200 that had a higher transcript level in response to 7-day LN treatments, suggesting that either the eccDNA stimulated the expression of the chromosomal gene or that the Os11g0634200 gene present in eccDNA molecules expressed itself. Another *ecGene*

that partially overlapped with *OsACP1* was more abundant in the 14-day LP treatments, wherein chromosomal gene expression also exhibited an up-regulation. Therefore, the potential role of eccDNAs harboring complete or partial genes on the positive regulation of gene expression should also be considered. Strategies for analyzing transcripts spanning junction sites must be developed to distinguish whether they originate from chromosomal genes or eccDNAs.

Additionally, systems for exploring how gene amplification driven by eccDNAs affects plant phenotype should be established for future research. The valuable example of EPSP synthase gene amplification conferring glyphosate resistance in *A. palmeri* highlights the genomic plasticity provided by eccDNAs that enable plants to adapt to various stresses[19]. Our work opens research avenues to understand how nutrient deficiency regulates gene expression, potentially shedding light on how certain species thrive in soils with low nutrient availability.

In conclusion, our study provides critical insights into the role of rice eccDNAs in plant biology. We identify potential mechanisms linking *ecGenes* and *ecTEs* to growth, development, and adaptive strategies, particularly under N- and P-limiting conditions (Fig. 8). Moreover, we developed robust statistical workflows to analyze differential elements within eccDNAs, uncovering significant genomic plasticity in rice. This research advances our understanding of rice eccDNAs and presents opportunities for further exploration of the regulation of plant responses to environmental stresses.

## Methods

### Plant material and genomic DNA isolation

Rice (*Oryza sativa* ssp. *japonica* cv. *Nipponbare*) was grown using a hydroponic system with modified Yoshida solution under controlled conditions (30 °C for 14 h during light/day and 26 °C for 10 h at dark/night). After pre-germination in tap water for 2 days, seeds were transferred to optimal nutrient solution containing 0.3125 mM $(NH_4)_2SO_4$, 0.3125 mM $Ca(NO_3)_2$, 0.1 mM $NaH_2PO_4$, 0.2565 mM $K_2SO_4$, 0.1865 mM $CaCl_2$, 0.8215 mM $MgSO_4$, 4.5 µM $MnCl_2$, 0.0375 µM $(NH_4)_6Mo_7O_{24}$, 9.5 µM $H_3BO_3$, 0.076 µM $ZnSO_4$, 0.0775 µM $CuSO_4$ and 0.018 mM EDTA-Fe. Seedlings were grown for 8 days, and the growth solution was renewed every 2 days before being transferred to nutrient stress treatments. For the low nitrogen (LN) treatment, seedlings were grown in solution with 0.0625 mM $(NH_4)_2SO_4$ and $Ca(NO_3)_2$ each for 3 days (short-term LN) and 7 days (long-term LN) treatments. For the low phosphorus (LP) treatments, seedlings were grown in a growth medium containing 5 µM phosphate ($NaH_2PO_4$) for 7 days (short-term LP) and 14 days (long-term LP). Nutrient solutions for both nutrient stress treatments and optimal conditions were renewed daily during the whole period. Two biological replicates of each group were performed strictly for harvesting shoot tissue to continue the subsequent experiments. Plants for tissue harvesting were selected randomly.

Genomic DNA (gDNA) from rice shoots was isolated using the CTAB method to prepare Oxford Nanopore sequencing templates[64]. RNAse A (omega BIO-TEK, Cat. No. AC118-1) treatment was required to maintain a higher quality gDNA. After the column purification with Genomic DNA Clean & Concentrator™-10 kit (ZYMO RESEARCH, Cat. No. D4011), gDNA quality and concentration were determined through agarose gel electrophoresis and Nanodrop™ One Spectrophotometer (Thermo Scientific™ Invitrogen™).

### Linear DNA digestion

5 µg of purified gDNA from each sample was treated with ATP-dependent PlasmidSafe DNase (Lucigen, Cat. No. E3110K) to remove linear DNAs following the midi-size reaction protocol. In brief, 10 µL of PlasmidSafe DNase and 10 µL of 10 mM ATP were added every 24 h. After 4 days, products were purified using the Genomic DNA Clean & Concentrator™-10 kit (ZYMO RESEARCH, Cat. No. D4011) and eluted with 24 µL of 0.3× TE Buffer. This was followed by a mini-size preparation with an additional 16 h of digestion. The products were

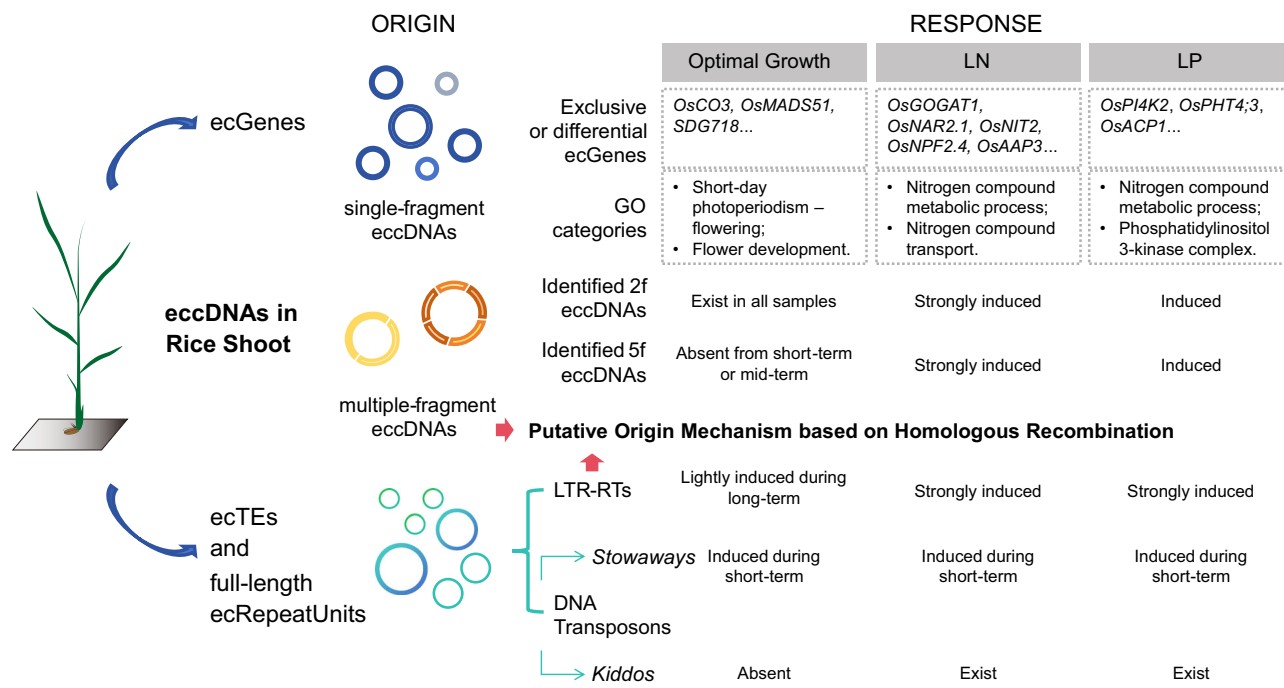

**Fig. 8 | Global eccDNA dynamics under optimal growth and nutritional stress conditions in rice shoot tissue.** eccDNAs are derived across the rice genome with single-fragment as the typical predominant class. Gene-overlapped eccDNAs (*ecGenes*), Transposable Element-overlapped eccDNAs (*ecTEs*), and full-length repeat unit-overlapped eccDNAs (*full-length ecRepeatUnits*) show dynamic changes during rice optimal growth, and in response to short-term and long-term low nitrogen (LN) and low phosphorus (LP) stresses. The accumulation of identified two-fragment and five-fragment eccDNAs (2 f and 5 f eccDNAs) is induced by nutritional stresses. Furthermore, a putative eccDNA origin mechanism based on Homologous Recombination (HR) among LTRs has been proposed by analyzing identified 2 f eccDNAs.

purified and eluted again. Then, it was followed by a mini-size preparation with 16 h of digestion. Finally, the pure eccDNAs, after 128 h of digestion, were concentrated in 15 µL of 0.3× TE Buffer for later enrichment.

### Enrichment of extrachromosomal circular DNAs
The random Rolling Circle Amplification (rRCA) was performed using the illustra™ TempliPhi 100 Amplification Kit (GE Healthcare Life Sciences, Cat. No. 25-6400-10) following the manufacturer's instructions. In brief, 0.5 µL eccDNA template was mixed with 5 µL Sample Buffer, and then heated at 95 °C for 3 min. After cooling down the template to 4 °C, 5 µL of Reaction Buffer and 0.2 µL of Enzyme Mix were added and incubated at 28 °C for 65 h for eccDNA amplification. After the enrichment, products were tested with *Eco*R1 digestion and compared to the gDNAs.

Debranching and polishing steps were required to prepare de-hyperbranched rRCA products for Oxford Nanopore sequencing. rRCA products were purified using magnetic beads (Omega BIO-TEK, Cat. No. M1378-00), and 200 µL samples from 5 rRCA reactions were mixed with 36 U DNA polymerase phi29 (New England Biolabs, Cat. No. M0269S), reaction buffer containing BSA, then incubated at 30 °C for 2 h. The reaction was inactivated by heating at 65 °C for 5 min. 200 U S1 nuclease (Thermo Scientific™ Invitrogen™, Cat. No. EN0321) digested the single-strand rRCA branches at 37 °C for 30 min, followed by magnetic bead purification. For gap filling and polishing, 12 U T4 DNA polymerase and 40 U DNA polymerase I (New England Biolabs, Cat. No. M0209S) were incubated with NEB buffer II at 25 °C for 1 h and inactivated at 75 °C for 10 min. The final debranched and polished products were purified using magnetic beads and quantitated using Quant-iT™ PicoGreen™ dsDNA Assay Kit (Thermo Scientific™ Invitrogen™, Cat. No. P7589), then quality controlled for library preparation to generate the library that was ready for sequencing.

### Sequencing and data analysis
Library preparation for the Oxford Nanopore sequencing platform was conducted using the Ligation Sequencing Kit (SQK-LSK110), following the manufacturer's instructions, and loaded into a PromethION flow cell (FLO-PRO002) for sequencing. After sequencing, raw data from each sample were basecalled using Guppy (version6.1.5). The parameters used for Guppy were as follows: --flowcell FLO-PRO002 --kit SQK-LSK110 --calib_detect --trim_barcodes --trim_strategy dna --disable_pings --device auto --num_callers 16. The output data in fastq format was further processed through Porechop (version0.2.4) to remove adapters. The parameters used for Porechop were as follows: --format fastq --extra_end_trim 0 --discard_middle. Clean data were then quality control analyzed using LongQC (version1.2.0c) to be ready for eccDNA identification. The parameters used were the following: sampleqc --x ont-ligation. LongQC results on raw data and clean data are presented in Supplementary Data 1.

### Identification and distribution analysis of eccDNAs
ecc_finder[27] (version1.0.0) (https://github.com/njaupan/ecc_finder) was used to identify eccDNAs, with the chromosomal rice genome as the reference. This tool excludes long reads originating from linear genomic repeats and removes alignments shorter than 200 bp with fewer than two repeat units, or those with a divergence rate exceeding 25% between repeat units. Finally, it retains loci covered by at least three reads, with a minimum of 80% of read length aligning to the region where eccDNAs originate. This ensures high accuracy and efficiency in plant eccDNA identification from nanopore sequencing data. The chromosomal distribution of eccDNAs across the whole genome was plotted through karyoploteR (version1.20.3). The density of eccDNAs per 100 kb was calculated through samtools (version1.12) and bedtools (version2.30.0), then plotted through circlize (version0.4.16). The size distribution of eccDNAs, together with the other visualizations, were plotted through ggplot2 (version3.3).

## Differential *ecGenes* analysis

The overlap between genes and eccDNAs was identified using bedtools (version2.30.0), with eccDNAs overlapping genes designated as *ecGenes*. A counting matrix of *ecGenes* detected in each treatment was constructed based on reads number for differential analysis. The count data exhibited a zero-inflated *poisson* distribution, as determined using the performance package (version 0.10.9) in R. Given this distribution, the GMPR[65] function from GUniFrac package (version1.8) was applied to normalize the counting matrix. Differential analysis of *ecGenes* was performed using EMDomics[66] (version 2.24.0) following its instructions, considering *ecGenes* with a *q*-value below 0.05 as differentially enriched. Differentially enriched *ecGenes* present exclusively in both replicates of a single treatment were classified as exclusive *ecGenes*, while those with higher read numbers in one treatment compared to another were categorized as differential *ecGenes*. Visualizations of specific differential and exclusive *ecGenes* were generated using karyoploteR and DNAMAN (version10.2.108).

## Gene Ontology enrichment and REVIGO analysis

GO enrichment analysis was performed for differentially enriched *ecGenes* and exclusive *ecGenes* through the web page supported by PANTHER[67] (http://geneontology.org). To reduce highly redundant categories and make a more accurate interpretation, both *p-values* from the GO enrichment analysis, together with the IDs of GO categories, were used for REVIGO[68] summary through the online program REVIGO (version 1.8.1) (http://revigo.irb.hr). The filtering settings were adjusted to "Medium (pin=0.7)" as recommended[68]. The resulting lists from REVIGO were visualized as dot plots using ggplot2.

## RNA-seq analysis

Rice shoot tissue from all treatments, with three biological replicates each, was used to isolate total RNA for RNA-seq. Plants for tissue harvesting were selected randomly. Fastp (version0.23.2) was utilized for trimming adapters and filtering low-quality reads to generate clean data. Pseudo-alignments were performed using Kallisto (version0.50.1) quantification. The parameters used for Kallisto were as follows: kallisto quant -i index -o output -b 150 -t 4. After performing the function of "abundance estimates to matrix" from the Trinity RNA-seq (version2.15.1) assembly software package with the parameters as --est_method kallisto --gene_trans_map none --out_prefix output --name_sample_by_basedir, the gene expression matrix has been created from kallisto abundance files and rounding adjusted for the differential statistical analysis through default setting of DESeq2 (version1.34.0). All visualizations were done with R packages, including ggplot2 and pheatmap (version1.0.12).

## Transposable elements and repeat units analysis

The rice TEs database was annotated using TransposonUltimate[69] and downloaded from the O_SATIVA_JAP folder named Annotation on the web page (https://cellgeni.cog.sanger.ac.uk/browser.html?shared=transposonultimate/). TransposonUltimate, a newer and well-developed tool for TE detection and classification in rice, provides more comprehensive information on the classification. The overlap analysis between TEs and eccDNAs was done by bedtools. eccDNAs that overlapped with TEs were named as *ecTEs*. Based on the classification information from TransposonUltimate[69], *ecTEs* were sorted into DNA transposons and Retrotransposons. Then, the detailed categories, for instance, *Zator*, *hAT*, and *Tc1-Mariner* from DNA transposon, as well as *Gypsy* and *Copia* from Long Terminal Repeat (LRT) retrotransposon, were described and visualized for each treatment through ggplot2.

To gain more information on the unique family of full-length repeat units that are associated with rice eccDNAs, the overlap between the repeat unit database (downloaded from RAP-DB[70] https://rapdb.dna.affrc.go.jp/download/irgsp1.html) and eccDNAs was done

with bedtools. Unique eccDNAs that fully cover the repeat units were named *full-length ecRepeatUnits*. Besides LTRs, DNA transposon families, for instance, *Stowaway* and *Kiddo*, were taken to the Sankey diagram using the sankeyNetwork function from the networkD3 package (version0.4) in R. The visualizations of specific *full-length ecRepeatUnits* were plotted through karyoploteR.

## Multiple-fragment eccDNA characterization

eccDNA_RCA_Nanopore[11] (https://github.com/icebert/eccDNA_RCA_nanopore) was utilized to identify multiple-fragment eccDNAs (MF-eccDNAs) using rice chromosomal genome as a reference. After applying the filters for the parameters "Nfullpass" and "Nfragment," both set to 1, the resulting eccDNAs were classified as MF-eccDNAs. By continuing the overlap with the gene reference file through bedtools, MF-eccDNAs were further grouped based on the detected core genes to count read numbers and reduce redundancy. The chromosomal distribution and visualizations on unique MF-eccDNAs were plotted through karyoploteR and DNAMAN. The statistical analysis of MF-eccDNAs was represented through the use of ggplot2.

## Inverse PCR validation

Genomic DNA was extracted from rice shoot tissue using the CTAB method. After digestion with ATP-dependent PlasmidSafe DNase (Lucigen, Cat. No. E3110K) for 128 h, rRCA was performed and purified with Genomic DNA Clean & Concentrator™-10 kit (ZYMO RESEARCH, Cat. No. D4011). gDNA, pure eccDNA products following PlasmidSafe DNase digestion, and the rRCA products were used as the templates for inverse PCR validations. Inverse primers were designed and blasted using NCBI's online tool: https://blast.ncbi.nlm.nih.gov/Blast.cgi. Both 2x Phanta Max Master Mix (Vazyme, Cat. No. P515-01) and Taq DNA Polymerase with ThermoPol® Buffer (New England Biolabs, Cat. No. M0267S) were utilized in PCR validation following their official recommendations. PCR products were checked through agarose gel electrophoresis first and then purified using E.Z.N.A.® Gel Extraction Kit (Omega BIO-TEK, Cat. No. D2500-01) for further validations of sequences from SANGERseq, KBseq based on the Illumina platform, or the Whole Plasmid Sequencing based on the ONT. The visualizations of specific sequencing fragments were plotted through karyoploteR and DNAMAN.

## ATAC-seq and eccDNA validation

ATAC-seq was conducted on two biological replicates of rice samples subjected to three different conditions: 7-day control samples (*Ctrl_D7*), 7-day low nitrogen treatment (*LN_D7*), and 7-day low phosphorus treatments (*LP_D7*), to further validate the presence of eccDNAs in rice. Plants for tissue harvesting were selected randomly. Nuclei Isolation Buffer (NIB), PBS with BSA, and PBSBR (PBS with BSA and RNAse inhibitor), and 60% Percoll solution were prepared then pre-cooled for nuclei isolation. -0.2 g of rice shoot tissue was chopped in 2 mL of NIB and filtered through a 40 μm cell strainer. The nuclear suspension was centrifuged at 4 °C, 2000 g, for 15 min. The nuclei pellet was washed with PBS, centrifuged again, and then washed with 60% Percoll solution. After washing with Percoll, the suspension was centrifuged at 4 °C, 1000 g for 10 min. The nuclei pellet was washed twice with PBSBR and resuspended in 200 μL of PBSBR. The nuclear suspension was quantified, and 10^6 nuclei were used for subsequent Tn5 transposase digestion. Following the manufacturer's instructions for the TruePrep® DNA Library Prep Kit V2 for Illumina (Vazyme, Cat. No. TD501), the nuclei were digested with Tn5 transposase. The resulting products were purified, followed by PCR to add adapters. Fragments were size-selected between 250 bp and 1000 bp using magnetic beads. The quality-controlled ATAC-seq libraries were then prepared for sequencing on the Illumina Novaseq6000 platform. ATAC-seq raw data were adapter-trimmed, and low-quality reads were filtered through fastp (version0.23.2), followed by quality control with

FastQC (version0.11.5). FastQC results on raw data and clean data were obtained in Supplementary Data 15. Clean data were further analyzed using the short-read pipeline from ecc_finder[45] with chromosomal genome as a reference for detecting high-density regions of eccDNAs in rice.

## Statistics and reproducibility
Each section of the Methods provides details on the study design, sample size, and statistical analyses of the data. For classic ATACseq, all procedures were done by two graduate students who were not aware of the expected results concerning eccDNAs (M.C., L.G., co-authors), from hydroponic planting to library preparation. All sequencing was done by the scientific-technical team, who knew nothing related to eccDNA analysis. Other experiments have no requirements for blinding. No data were excluded from the analysis.

## Reporting summary
Further information on research design is available in the Nature Portfolio Reporting Summary linked to this article.

## Data availability
The raw sequencing data generated in this study have been deposited at the National Center for Biotechnology Information (NCBI) under Bioproject accession-ID: PRJNA1136237. Source data are provided with this paper.

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

## Acknowledgements

This study was partially supported by the National Key Research and Development Program of China (2021YFF1000400), Jiangsu Seed Industry Revitalization Project (JBGS [2021] 011) to G.X., and State of Texas Governor's University Research Initiative (GURI)/ Texas Tech University (grant 05-2018) to L.H.-E. We thank the technical support from the high-performance computing platform of Bioinformatics Center, Nanjing Agricultural University. We thank Zhitao Zhu (Bioinformatics Center, Nanjing Agricultural University) for his outstanding assistance in computing platform testing. We thank Shichao Wang member of the National Key Laboratory of Crop Genetics & Germplasm Enhancement and Utilization, Nanjing Agricultural University, and Gerardo Alejo-Jacuinde, Francisco Pérez-Zavala, Gabriela Cabrales-Orona, Benjamin Pérez-Sánchez, and Valeria Flores-Tinoco, members of the Institute of Genomics for Crop Abiotic Stress Tolerance, Texas Tech University, for their invaluable advice on this research.

## Author contributions

Experimental design: H.N., L.Y.-V., G.X., and L.H.-E.; eccDNAs extraction, detection, and validation: H.N., L.Y.-V., and M.G.; Nuclear extraction and ATAC-seq: M.C., L.G., L.Y.-V., and M.G.; data analysis: H.N., L.Y.-V., D.L.-

A., G.X., and L.H.-E.; manuscript writing and revision: H.N., L.Y.-V., D.L.-A., G.X., and L.H.-E. All authors read and approved the manuscript.

## Competing interests

The authors declare no competing interests.
