## [Transparent Peer Review file · Nature Communications]

Adaptive Dynamics of Extrachromosomal Circular DNA in Rice Under Nutrient Stress

Corresponding Author: Professor Luis Herrera-Estrella

Version 0:

Reviewer comments:

Reviewer #1

(Remarks to the Author)

The manuscript by Ni and colleagues reports on the characterization of extrachromosomal circular DNA (eccDNA) in rice upon different nutrient deficiencies (nitrogen and phosphorus). eccDNA-seq was performed on shoot tissue. The authors identified a high number of eccDNAs corresponding to genes and Transposable elements (TEs), some of them being affected by the treatment.

I enjoyed reading the paper, it is clearly written and the data is nicely presented. I have questions concerning both the visualisation and interpretation of the results.

1) As a general remark it is difficult to appreciate the length of the eccDNA for the lack of visual representation of the raw data (ONT reads). For instance on IGV views on Fig 2e and 3e only the position of the ecc_finder candidates are shown (bed file), not the raw reads. It is informative to see the reads to better appreciate the coverage and specificity of the eccDNA-seq signal. Only Supp Fig 13 shows the reads, and here the total coverage would be helpful. Also a general coverage plot (e.g. samplot) can help visualising the data. Another type of data that could be useful is a dot plot showing the different repeated units present in selected ONT reads and corresponding to the RCA step. The authors focus on GO enrichment but it could be informative to go more into the details of a few well-characterized eccDNAs, as this topic is relatively new and not familiar to all potential readers.

2) Concerning the interpretation: while the authors nicely describe the response of the eccDNA repertoire upon stress treatment, a mechanism is not yet in hand. For example how would a partial gene found as eccDNA interfere with gene expression is, so far, not understood. The discussion could clarify the speculative parts concerning the mechanisms in place. For example line 552-555: although interesting, this seems correlative at this point. Idem for line 576 and 616.

3) On chloroplast sequences: Line 272. During RCA the chloroplast and mitochondrial genomes are amplified and usually represent a large fraction of eccDNA-seq data. How can the authors exclude that these ecGenes are not from the chloroplast genome? Some other highly amplified regions could also be checked (line 318, line 415).

4) On ecGenes (eccDNAs bearing genes): are genes represented in their full length form in eccDNA or only partially? In the exemple detailed in Fig 2e for instance, it could be that the first intron of the gene contains an unannotated TE forming eccDNAs. Is this intron sequence unique in the genome? On Fig 3a/b: What is the overlap between ecGenes in D3 versus D7?

5) On ecTEs: it seems surprising that 611 TEs could be active and produce eccDNA in rice. The data on Stowaway and Kiddo seems very exciting and would deserve to be described in more details (see point 1).

6) In general, across the different treatments, how do the two biological replicates overlap?

Minor points:

Line 141: Transposable elements (TEs)

Fig 2f: why is the size different between gDNA and RCA?

Fig 6: it could help if the genes function and sample of origin (what treatment) was indicated in the Figure (d/e).

Line 536: it seems to be the other way around: a role for DNA methylation on rice ecTEs.

Some italics are missing (Tc1 Mariner, Copia etc).

Reviewer #2

(Remarks to the Author)

The manuscript by Ni et al. reported the identification of extrachromosomal circular DNAs (eccDNAs) in rice under N/P treatments. Given the important role eccDNAs in modulating the genomic plasticity, this work will increase our understanding toward the diverse regulatory mechanisms of nutrient response in plants. This manuscript provides substantial data in exploring the N/P responsive eccDNAs, and also gives an interesting working model for illustrating the involvement of eccDNAs in nutrient response. However, to obtain a solid conclusion in connecting the function of eccDNAs and nutrient response, several concerns still need to be addressed.

1. My major concern is about the supposed roles of these identified eccDNAs. As the authors mentioned, the average size of the eccDNAs identified in this study was approximately 500 bp (line 133). To our knowledge, such a small DNA fragment cannot work as an independent unit to start the gene expression or obtain a complete CDS. Thus, it might result in a misleading conclusion by analyzing all the eccDNAs without filtering the functional candidates. In my opinion, among the identified eccDNAs, there should exist the members with the complete gene structure. These members should be selected and analyzed firstly. For the members with very small size, as they cannot produce the full-length transcripts, it is necessary to explore their possible roles, possibly working as the regulatory elements to target the gene in the genome, the function of which can indicate the regulatory roles of the eccDNAs but not the eccDNA-mapped gene in the genome.

2. The authors identified several eccDNAs that are involved in N/P utilization. To support the proposed model, the validation of these eccDNA in producing the transcript of these N/P-utilized genes would be greatly helpful.

3. This study used the shoot to isolate the ccDNAs in response to N/P treatments. Why not roots?

Version 1:

Reviewer comments:

Reviewer #2

(Remarks to the Author)

In the revised manuscript, most of my concerns are properly addressed, and the manuscript is greatly improved. I have no further questions.

Reviewer #3

(Remarks to the Author)

This study investigates the dynamic changes of eccDNAs in rice across different growth stages and under nutrient stress conditions (nitrogen and phosphorus), focusing on their potential roles in plant growth, development, and adaptation to environmental stresses. Using Nanopore sequencing, RNA-seq, and ATAC-seq technologies combined with bioinformatics analysis, the authors systematically analyzed the size, origin, distribution, and categories of rice eccDNAs, as well as their association with gene expression, transposable elements (TEs), and potential roles in response to nitrogen and phosphorus deficiency. Overall, the paper highlights the significant role of rice eccDNAs in plant development and environmental response, providing new insights into the plasticity of plant genomes.

Despite the research value of this study, there are some issues that require further revision.

Q1. Line 125: Where are the evaluation results for these eccDNA identification tools (CIDER-Seq2, ECCsplorer, ecc_caller, and ecc_finder)? In particular, regarding CIDER-Seq2, can its input dataset be compatible with the Nanopore sequencing data format used in this study? In addition, I noticed another published software for eccDNA identification (FLED), which appears to offer higher efficiency and accuracy in eccDNA detection (<https://github.com/Fuyuli/FLED>). I recommend the authors take it into consideration.

Q2. Line 147: Given that the average size of eccDNA is 500 bp, is it reasonable to merge sequences with an overlap of more than 200 bp? If the overlapping regions are merely shared sequences without representing identical functional regions, over-merging could lead to misinterpretations. Relying solely on overlapping to merge could result in the loss or misjudgment of information. In certain cases, eccDNAs may show significant variations due to transposable element activity or DNA rearrangement, and these variants might exhibit long overlaps. If merging is entirely based on a 200 bp overlap, some meaningful variations or diversity in fragments might be overlooked. The discussion should address the potential problems

and limitations of this merging approach.

Q3. Line 151: The definition and limitations of ecGenes. When ecGenes contain complete genes, how can it be distinguished whether gene expression originates from genomic amplification or eccDNA amplification? When ecGenes include only partial fragments of genes, how can the causal relationship between changes in gene expression and ecGenes be determined?

Q4. Line 185: Regarding the classification of differentially enriched ecGenes as "exclusive," some limitations should be addressed: 1) Insufficient sequencing depth: If sequencing depth is inadequate, especially for low-abundance eccDNAs, the expression of certain genes may not be fully captured. This might lead to false negatives for genes that should exhibit significant differences. Conversely, genes with fewer reads in one treatment group may be mistakenly identified as differentially expressed due to sequencing errors. 2) Lack of biological context and functional validation: While using q-values and biological replicates to identify differentially expressed genes is reasonable, without additional functional validation or biological context, these genes might only represent statistical differences without real biological significance. These all need to be explained and clarified in detail during the discussion.

Q5. Line 318: About the issue of the relationship between RNA-seq and ecGenes. Although the RNA-seq analysis revealed a correlation between some ecGenes and chromosomal gene expression, this remains an association rather than a causative relationship. For example, while the authors note that certain ecGenes are associated with upregulated or downregulated chromosomal genes, it remains unclear whether these ecGenes directly participate in gene expression regulation. Further functional validation experiments are needed to establish causality.

Q6. For the Nanopore sequencing and ATAC-seq data, it is recommended to provide more detailed statistics on sequencing depth and library saturation analysis, including a clear explanation of how biological replicates were managed for each treatment to minimize noise and validate the robustness of the observed differences. While the authors mention LongQC in the Methods section, additional details on the actual average coverage depth, duplication rates, and replicate correlations should be included in the Supplementary Data.

Q7. When identifying differentially enriched eccDNAs, particularly differential ecGenes, the authors used GMPR normalization and EMDomics, which is conceptually feasible. However, given that zero counts of eccDNA are very common, have the authors considered other zero-inflated models (e.g., ZINB, etc.) for comparison? Alternatively, can additional sensitivity analyses be conducted from a methodological standpoint? Presenting the consistency of differential results under different normalization/statistical models would be more convincing.

Q8. The authors performed GO enrichment analysis for the differential ecGenes and used REVIGO to consolidate the results. However, if only part of a gene (e.g., partial CDS/UTR) is covered, it is not guaranteed that the function of that segment is the same as the full-length gene. It is recommended that the authors make a clearer distinction between "partial gene overlap" vs. "full-length overlap" in the Results and Discussion sections to avoid over-interpretation.

Q9. The article discusses in detail the potential activity of MITEs such as Stowaway and Kiddo, as well as LTRs, which is a novel highlight of this study. It is recommended that the authors include more explanation in discussion on the criteria for judging TE activity. For example, is there any evidence indicating that these elements indeed transpose or affect the expression of nearby genes under stress (e.g., changes in RNA-seq expression upstream or downstream of insertion sites)? The terms "active" or "potentially active" should be used more cautiously.

Q10. Some minor errors should be corrected, e.g., in Line 153, it should be written as 30.3% instead of simply 30%; similarly, in Line 157, it should be written as 8.0% instead of simply 8%. The entire text should be carefully proofread.

In conclusion, this study offers valuable insights into the potential roles of rice eccDNAs in plant development, particularly in environmental stress response, and proposes new hypotheses and research directions. However, the manuscript has some weaknesses in data analysis depth and exploration of biological mechanisms. To enhance the quality of the article, I recommend the authors address the aforementioned comments, providing additional reliable evidence.

Reviewer #4

(Remarks to the Author)

The authors have adequately addressed the reviewers' concerns. The revised manuscript has undergone significant improvements, which is now a valuable contribution to eccDNA research. However, I have two comments regarding the selection of analysis tools.

Major issues:

1. In the ecc_finder paper [1], the authors state, "We have not tested ecc_finder on ATAC-seq data, but a recent method for this specific type of data has been described." The specific algorithm for ATAC-seq data analysis is Circle_finder[2]. Why did the authors opt to use ecc_finder rather than Circle_finder for the ATAC-seq data analysis presented in Figure 7?
2. The authors applied eccDNA_RCA_nanopore to identify multiple-fragment eccDNAs. However, a recent benchmark study[3] indicates that CReSIL[4] outperforms eccDNA_RCA_nanopore in identifying eccDNAs from Nanopore data. Additionally, the identification results using eccDNA_RCA_nanopore exhibit redundancies. Therefore, I recommend that the authors consider using CReSIL for the analysis of multiple-fragment eccDNAs to validate their results.

References:

- [1] Zhang, P., Peng, H., Llauro, C., Bucher, E. & Mirouze, M. ecc_finder: A Robust and Accurate Tool for Detecting Extrachromosomal Circular DNA From Sequencing Data. *Front. Plant Sci.* 12, 743742 (2021)
- [2] Kumar, P. et al. ATAC-seq identifies thousands of extrachromosomal circular DNA in cancer and cell lines. *Sci. Adv.* 6, eaba2489 (2020).
- [3] Gao, X. et al. Comparative analysis of methodologies for detecting extrachromosomal circular DNA. *Nat. Commun.* 15, 9208 (2024).
- [4] Wanchai, V. et al. CReSIL: accurate identification of extrachromosomal circular DNA from long-read sequences. *Brief Bioinform.* 23 (2022).

Addressing these points will strengthen the manuscript, ensuring that it provides a comprehensive, accurate, and clear resource for readers interested in eccDNA research.

Version 2:

Reviewer comments:

Reviewer #3

(Remarks to the Author)

Overall Recommendation:

Accept after minor revisions (English language polishing)

Major Concerns Addressed:

The authors have made satisfactory revisions in response to the previous round of review comments, and the manuscript now aligns with the scientific standards of Nature Communications. However, several issues in English writing persist and require careful polishing to ensure clarity and professionalism.

Specific Revisions Required:

There are several inconsistencies in subject-verb agreement throughout the manuscript. For instance, at line 163, "of which 267 covered full-length gene" should be revised to "of which 267 covered full-length genes."; similarly, at line 207, 223, 236, 288, 306, 367, 383 etc. Additionally, minor spelling errors were noted. One example is at line 670, where "segment" was misspelled as "sagment". These and other similar errors should be carefully revised throughout the text. A professional proofreading service or a native English-speaking co-author should review the manuscript to address these issues.

Reviewer #4

(Remarks to the Author)

My concerns are adequately addressed in the revised manuscript. I have no further questions

Response to Reviewers' and Editorial Comments

We thank the editor for considering our manuscript and allowing us to revise it. The valuable feedback and suggestions have significantly improved our study's clarity and rigor.

As requested by the referees, we adjust the visual representation for each figure, clarify all experimental details on materials and methods, and perform additional analysis with discussion to respond to reviewers' questions and concerns on organelles, *ecGenes*, and rates.

To improve the strength of the study, we perform additional analysis by supplying new RNA-Seq data of plants subjected to N and P stress to have a more direct correlation between the presence of eccDNAs and changes in expression of specific genes, which helps interpretation of the impact of eccDNAs on chromosomal gene expression. As requested by both referees and editor, we clarify and expanded our models on the mechanisms of the formation and effect on gene expression of eccDNAs in a new discussion section.

To comply with the additional analysis requested by the two referees and editor, we revised the data presented in four figures (Figure 2-e,f; Figure 3-e; Figure 4-d,e; Figure 6-g,j,k) and included nine new supplementary figures (Supplementary Fig.5c-f; Supplementary Fig.8; Supplementary Fig.9c-e; Supplementary Fig.11; Supplementary Fig.12c-e; Supplementary Fig.14; Supplementary Fig.15d-f; Supplementary Fig.16b; Supplementary Fig.17f-g).

Below, we respond to each of the referees' comments.

Point by point response to Referee #1:

1) As a general remark it is difficult to appreciate the length of the eccDNA for the lack of visual representation of the raw data (ONT reads). For instance, on IGV views in Fig 2e and 3e, only the position of the ecc_finder candidates is shown (bed file), not the raw reads. It is informative to see the reads to better appreciate the coverage and specificity of the eccDNA-seq signal. Only Supp Fig 13 shows the reads, and here the total coverage would be helpful. Also a general coverage plot (e.g. samplot) can help visualising the data. Another type of data that could be useful is a dot plot showing the different repeated units present in selected ONT reads and corresponding to the RCA step. The authors focus on GO enrichment but it could be informative to go more into the details of a few well-characterized eccDNAs, as this topic is relatively new and not familiar to all potential readers.

Response

We thank the reviewer for these invaluable suggestions on improving the visual representation through ONT coverage and dot plots on ONT reads.

We replace all the IGV views with karyoplots of bam coverage from ONT data in the primary and supplementary figures: 1) Fig2-e and Fig3-e are replaced with new karyoplot including ONT coverage; 2) Fig4-d is updated to including ONT coverage; 3) Supplementary Fig17-f,g (used to be Supp Fig 13-f,g) are replaced with new karyoplot including coverage information. We also update other figures that need to be showed together with the ONT coverage (Supplementary Fig8-d; Supplementary Fig11-e,f; Supplementary Fig14-e).

For a more detailed visualization of validated eccDNAs, we include Dot plots of ONT reads or validation results in primary and supplementary figures: 1) Fig4-e is added as dot plot between KBseq results and reference sequences; 2) Fig6-g is added as dot plot between ONT read and reference sequences and Fig6-j,k are added as dot plots between Whole Plasmid Sequencing results and reference sequences; 3) Supplementary Fig8-a,b,c,g; Supplementary Fig9-e; and Supplementary Fig12-e are added as dot plots between ONT reads and reference sequences. We also added dot plots between ONT reads and reference sequences as requested by this reviewer (Supplementary Fig11-g, h; Supplementary Fig14-f).

To go more into the details of a few eccDNAs, we update the karyoplots (Fig2-e; Fig3-e; Fig4-d) on validated *ecGenes* of plants grown in optimal nutrient conditions (lines 247-248), LN (lines 318-321) and LP stresses (lines 402-405) to include SANGERseq or KBseq results. We add dot plot (Fig4-e; Supplementary Fig8; Supplementary Fig9-e; Supplementary Fig12-e) including ONT reads or KBseq results to show detailed looping structures corresponding to the RCA step (lines 251-254; lines 260-262; lines 321-323; lines 408-410). We also supply additional chromosomal gene expression levels on some *ecGenes* to show the association between *ecGenes* and chromosomal gene expression, which improves the justification for performing GO analysis with exclusive *ecGenes* (Supplementary Fig11-i; Supplementary Fig14-g). Considering the few functional plant eccDNAs that have been characterized in detail, such as the EPSPS amplicons, and GO analyses reported in previous studies (reference 15, reference 17), we have expanded the discussion (lines 752-758) on this topic to provide more clarity for our potential readers.

2) Concerning the interpretation: while the authors nicely describe the response of the eccDNA repertoire upon stress treatment, a mechanism is not yet in hand. For example how would a partial gene found as eccDNA interfere with gene expression is, so far, not understood. The discussion could clarify the speculative parts concerning the mechanisms in place. For example line 552-555: although interesting, this seems correlative at this point. Idem for line 576 and 616.

Response

We agree with the reviewer that a more detailed discussion should be included in the revised manuscript regarding the mechanisms by which eccDNA can impact gene expression.

To explore the potential mechanisms on rice eccDNAs that overlapped with gene fragments, we performed RNA-seq analysis of LN and LP treatments and then determined the association analysis between chromosomal DEGs and exclusive *ecGenes* (lines 324-343 and lines 412-426; Supplementary Fig11 and Supplementary Fig14). In several cases, amplification of incomplete genes in eccDNA molecules correlated with a reduction in the transcript level of the corresponding chromosomal gene, suggesting that siRNAs might be responsible for a decrease in gene expression. In a few other cases where complete genes were present in eccDNA molecules, gene expression increased, suggesting that the amplification of complete genes leads to increased expression

We further discuss our analysis together with a couple of previous studies (reference 10 and reference 17) in a new subsection of the discussion entitled “Potential functions of *ecGenes* in regulating chromosomal gene expression” (lines 727-761). Our analysis revealed a complex

association between *ecGenes* and the expression of their chromosomal counterparts, within the association that 50% of gene IDs that increased in the *ecGenes* from samples of LN and short-term LP treatments were identified as down-regulated DEGs. This result suggests a putative negative role of eccDNAs in the expression of chromosomal genes. Other than this, we also find a positive association between exclusive *ecGenes* and chromosomal DEGs, which suggests the potential role of eccDNAs harboring complete on the positive regulation of gene expression should also be addressed in the future.

To enhance the paper's structure and logic, we have provided a more detailed discussion on the effect of eccDNA on gene expression in the Introduction section (lines 60-62, reference 10). Additionally, we restructured the relevant part of the Discussion to clarify the role of eccDNAs that overlap with functional genes during LN and LP treatments. The subsection previously spanning lines 552-616 has now been expanded and renamed to 'eccDNAs Overlapping with Functional Genes in Rice Growth and Responses to N and P Stresses' for better distinction and clarity.

3) *On chloroplast sequences: Line 272. During RCA the chloroplast and mitochondrial genomes are amplified and usually represent a large fraction of eccDNA-seq data. How can the authors exclude that these ecGenes are not from the chloroplast genome? Some other highly amplified regions could also be checked (line 318, line 415).*

Response

This is an important observation from the reviewer. First, we clarify in the revised version of methods (lines 844 and line 921) and the main text (lines 134-135; lines 500-501) that for the analysis of long-reads using *ecc_finder* and *eccDNA_RCA_Nanopore* only sequencing reads that mapped to the chromosomal genome were used for the downstream analysis. Therefore, sequences from organellar genomes were excluded.

Since in the rice nuclear genome, there are regions derived from the mitochondrial (NUMTs) and plastid genomes (NUPTs), we determine the proportion of eccDNAs that originated from these regions (Supplementary Fig5-c, d, e, f). These results are described in lines 169-178 of the revised manuscript, showing that among all IDed-eccDNAs, less than 1.3% originated from NUMTs or NUPTs. Our results are further discussed together with the study on rice eccDNAs using short Illumina reads that revealed only around 0.5% of eccDNAs in rice were derived from the MT or PT genomes (Zhuang, *et al.* 2024) (lines 174-175). We conclude that eccDNAs are distributed widely across the rice chromosomal genome (line 180).

Finally, we double-check the exclusive *ecGenes* characterized for each treatment to see if any were derived from nuclear regions with homology to the mitochondrial (NUMTs) or plastid genome (NUPTs) regions. Only 0.4% to 0.7% of *ecGenes* had homology to NUMTs or NUPTs regions. Excluding these *ecGenes* did not change the REVIGO results.

Considering the suggestion that “*Some other highly amplified regions could also be checked,*” we carefully analyzed all highly amplified regions in our manuscript. We find that MF-eccDNAs with Os05g0372300, Os04g0343050, and Os01g0791033 as core genes had a fragment with homology to NUMTs, but no fragments with homology to the plastid genomes. The mitochondrial genome can originate subgenomic circular DNA in plants through homologous recombination events. However, we found no evidence that the NUMT fragments

in MF-eccDNAs were derived from subgenomic mitochondrial circular DNA. The NUMT fragments present in MF-eccDNAs contained some deletions compared to the corresponding NUMT regions. Therefore, we are more inclined to classify these MF-eccDNAs as of nuclear rather than organellar origin. We hypothesize that MF-eccDNA could contain some fragments of organellar origin and could be the source of NUMT and NUPT regions in plant genomes, supported by the evidence of eccDNA intracellular mobility shown in mammalian cell lines (reference 11).

4) *On ecGenes (eccDNAs bearing genes): are genes represented in their full length form in eccDNA or only partially? In the example detailed in Fig 2e for instance, it could be that the first intron of the gene contains an unannotated TE forming eccDNAs. Is this intron sequence unique in the genome? On Fig 3a/b: What is the overlap between ecGenes in D3 versus D7?*

Response

Most *ecGenes* contained partial fragments of their chromosomal genes (line 156).

We double-check the validation of the example presented in *Fig2-e* in our previous manuscript and find that *ecGene* originated from the last intron that covered a set of the same repeat units annotated as *ORSgTEMTO1602424* (new karyoplot was added as Supplementary Fig8-d). We further blast this sequence and find it is not unique in the rice genome. A detailed description is now included in the revised version of our main text (lines 255-262), and all validations together with a dot plot on this *ecGene* are moved and added into supplementary Fig8.

Instead, we include a new validation of *ecGene* (Os10g0439924) present in both replicates of *Ctrl_D14* (the karyoplot and gel electrophoresis were replaced with revised Fig2-e, f; dot plots showing ONT reads were added as Supplementary Fig8-a, b,c). A detailed description of this *ecGene* is presented in the revised version of our manuscript (lines 247-255).

We also include additional analysis of the overlaps between exclusive *ecGenes* in the two nutrient deficiency treatments, which was pointed out in *Fig 3a/b (D3 versus D7)*. The overlaps between exclusive *ecGenes* in *LN_D3* and *LN_D7* are 104, while in *Ctrl_D3* and *Ctrl_D7* are 8. Venn diagrams of these analyses are presented in Supplementary Fig9-c, d and under LP treatments in Supplementary Fig12-c,d. These new results are now described in the revised version of our manuscript (lines 308-311; lines 387-390).

5) *On ecTEs: it seems surprising that 611 TEs could be active and produce eccDNA in rice. The data on Stowaway and Kiddo seems very exciting and would deserve to be described in more details (see point 1)*

Response

We thank the reviewer for this suggestion. Now, we describe the data on *Stowaway* and *Kiddo* elements in more detail. According to our analysis, 51.2% of 96757 IDed-eccDNAs were identified as overlapping to TE regions (Fig1-e), the high-density regions of eccDNA were linked with the genomic regions of high TE density (Fig1-c), and we found 611 ecTEs that were common to all treatments (Supplementary Fig15-b). Considering more than 45% of the rice genome is contributed by TEs, our findings suggest a role of TE activity in eccDNAs production.

Following the review's suggestion, we add a new analysis on *STOWAWAY11_OS* to clarify the evidence on the potential activity of *Stowaway- full-length ecRepeatUnits* in the revised version of our manuscript (lines 464-475). Karyoplot with bam coverage from raw ONT data is added as Supplementary Fig16-b. The detailed subfamilies of *Kiddo* are also added to our main text (lines 478-485). Karyoplots with bam coverage on all the *Kiddo- full-length ecRepeatUnits* are added as Supplementary Fig15-d, e, f. Both *full-length ecRepeatUnits* are further discussed in more detail in our revised manuscript (lines 624-632; lines 633-649).

6) In general, across the different treatments, how do the two biological replicates overlap?

Response

To answer the reviewer's concern, we provide the karyoplots for the two replicates in each treatment (including density / 100 kb and all eccDNA regions) and the PCA analysis between contrasts and treatments to show an overlap between two biological replicates. These results could be included in a supplementary figure if the reviewer finds it relevant to provide readers with this information.

(line chart shows the density of eccDNAs per 100 kb, bar chart shows the regions of all eccDNAs detected in this sample)

(PCA analysis was performed through labels based on IDed-eccDNAs. Only eccDNAs with an overlap of over 200 bp will be given the same label.)

Minor points:

Line 141: Transposable elements (TEs)

Response

All the spelling of Transposable elements is corrected in the main text.

Fig 2f: why is the size different between gDNA and RCA?

Response

During the design of primers for inverse PCR of the *ecGene* presented in Fig 2f of the original manuscript (now in supplementary Fig.8 in the revised manuscript), we tried to get more specific primers. However, due to the high similarity of the repeated sequences in this locus, non-specific amplification with larger fragments is observed in primer-BLAST results. Considering that the predicted products from gDNAs differed from the *ecGene* in size, we performed the SNAGERseq on the amplification products. Our analysis revealed that the products from gDNAs were not in our target region, which validated that the products from gDNAs were the nonspecific amplification on the chromosomes and in different fragment sizes.

Fig 6: it could help if the genes function and sample of origin (what treatment) was indicated in the Figure (d/e).

Response

Gene functions and sample origins are added to figure legends legends.

Line 536: it seems to be the other way around: a role for DNA methylation on rice ecTEs.

Response

Details on *Kiddo* and the hypothesis on rice *ecTEs* are revised in the Discussion (lines 633-640).

Some italics are missing (Tc1 Mariner, Copia etc).

Response

Italics are added to all TE families.

Point-by-point response to Referee #2:

1) My major concern is about the supposed roles of these identified eccDNAs. As the authors mentioned, the average size of the eccDNAs identified in this study was approximately 500 bp (line 133). To our knowledge, such a small DNA fragment cannot work as an independent unit to start the gene expression or obtain a complete CDS. Thus, it might result in a misleading conclusion by analyzing all the eccDNAs without filtering the functional candidates. In my opinion, among the identified eccDNAs, there should exist the members with the complete gene structure. These members should be selected and analyzed firstly. For the members with very small size, as they cannot produce the full-length transcripts, it is necessary to explore their possible roles, possibly working as the regulatory elements to target the gene in the genome, the function of which can indicate the regulatory roles of the eccDNAs but not the eccDNA-mapped gene in the genome.

Response

We fully understand the reviewer's concern.

According to our analysis, most *ecGenes* originated from CDS (Fig1-g) and were identified as partial gene fragments. Differential analysis through EMDomics showed the exclusive *ecGenes* through optimal growth or under nutritional stresses, which could help us focus more on the potential functional *ecGenes* responsive to nutrient limitations. Detailed analysis revealed only a few exclusive *ecGenes* obtained a full-length gene structure.

Considering these situations and to explore more evidence for the regulatory roles hypothesis suggested by the reviewer, we perform additional classic RNA-seq for all LN and LP treatments and then apply the association analysis between chromosomal DEGs and exclusive *ecGenes* to clarify putative roles for both partial *ecGenes* and full-length *ecGenes* (lines 324-343 and lines 412-426; Supplementary Fig11 and Supplementary Fig14).

Under LN stresses, we identify one full-length *ecGene* (Os12g0573600) whose frequency was increased in the LN treatment, but its expression was significantly down regulated. Another full-length *ecGene* (Os11g0634200) was identified as induced by LN treatment but with up-regulated expression. While under LP stresses, we validate one partial *ecGene* (*OsACPI*), which is abundance increases by LP treatment and still has an up-regulated expression. All this evidence suggests a complex association between *ecGenes*, no matter whether obtaining partial gene or full-length gene, and their expression levels.

We provide all these new analyses in our revised manuscript on lines 324-343 and lines 406-408, 412-426. Detailed figures are added to Supplementary Fig11 and Supplementary

Fig14. We further discuss our analysis together with previous studies in rice and mammalian cell lines (reference 10 and reference 17) in a new subsection, “Potential functions of *ecGenes* in regulating chromosomal gene expression” of Discussion (line: 727-761) and clarify that “Innovative strategies to determine whether transcripts are derived from chromosomal genes or those present in eccDNAs need to be developed to determine the potential role of gene amplification on plant phenotype.”

We explored the potential mechanisms behind the formation of *ecGenes* by analyzing all exclusive *ecGenes* to formulate mechanisms of eccDNA formation and address their complex role in rice genome plasticity. To clarify and avoid confusion, we have revised the subsection on the involvement of eccDNAs overlapping with genes during LN and LP treatments. Previously found in lines 552-616 of the Discussion, this content has been reorganized under a new subtitle: “eccDNAs Overlapping with Functional Genes in Rice Growth and Responses to N and P Stresses.”

2) The authors identified several eccDNAs that are involved in N/P utilization. To support the proposed model, the validation of these eccDNA in producing the transcript of these N/P-utilized genes would be greatly helpful.

Response

We thank the reviewer for this suggestion.

We perform additional classic RNA-seq as supplied in the first comment to investigate the chromosomal transcription level changes and look for more transcript evidence showed on those *ecGenes* that we validated under nutrient stresses. We find a positive association between the inducing of *ecGene* related to *OsACPI* (which codes a phosphate starvation-induced acid phosphatase) and its chromosomal gene expression levels under long-term LP treatment. Although this *ecGene* obtained its chromosomal gene partially, its chromosomal transcript was still significantly induced by LP stress, revealing a potential mechanism on the regulatory role of eccDNAs in response to phosphorus.

These results are presented in the revised manuscript, specifically in lines 402-411, with additional figures included in Supplementary Fig. 14. However, we acknowledge the limitations of classic RNA-seq, which cannot distinguish between the expression of genome-encoded genes and those located on eccDNA molecules. We further discuss the potential regulatory mechanisms by which eccDNAs regulate gene expression but also point out the limitations of current technologies to differentiate transcripts derived from *ecGenes* and chromosomal genes in the new subsection, 'Potential Functions of *ecGenes* in Regulating Chromosomal Gene Expression' (Discussion, lines 723-750). Additionally, we emphasize the need for new strategies in the future to differentiate transcripts originating from *ecGenes* and their genomic counterparts (lines 752-761)."

This revision improves the flow and makes the ideas more concise and more accessible to follow.

3) This study used the shoot to isolate the ccDNAs in response to N/P treatments. Why not roots?

Response

We used leaf tissue for five main reasons: 1) Nutrient deficiencies have more severe physiological and biochemical effects on the shoot than on the root; 2) Hydroponic culture allows for easier and more precise sampling of roots for eccDNA analysis compared to soil culture, 3) root responses to nutrient deficiency in liquid culture differ significantly from those in soil, while the difference in shoot responses between the two culture systems is much smaller; 4) Previous studies, including Zhuang *et al.* (2024) (reference 10), have shown that rice accumulates a higher diversity and greater quantity of eccDNAs in leaves than in roots. These points are now clearly stated in the revised version of our manuscript (lines 110-113).

This revision improves readability by simplifying the structure and ensuring the points are more clearly linked.

Response to Referee and Editorial Comments

We thank the editor for considering our manuscript and allowing us to revise it. The valuable feedback and suggestions from the referees have greatly improved the clarity and rigor of our study.

As requested by the referees, we test additional identification tools on eccDNAs, revise details in results, discussion, and figure legends and captions accordingly, correct minor errors, and clarify our experiment and data analysis processes in our manuscript's materials and methods section.

To meet the detailed supplementary data requested by the reviewers, we have revised and provided new supplementary data on Nanopore sequencing qualification control (QC) (Supplementary Data 1), ATAC-seq QC (Supplementary Data 15), multiple-fragment eccDNAs (Supplementary Data 14), and have additionally included new supplementary data on exclusive *ecGenes* covering the full-length gene during both optimal growth and under nutritional limitations (Supplementary Data 3, 7, 11).

Below, we respond to each of the referees' comments.

Response to Referee #2:

In the revised manuscript, most of my concerns are adequately addressed, and the manuscript has seen significant improvement. I have no further questions.

Response:

We thank our referee for the approval on our previous version of the manuscript.

Response to Referee #3:

This study investigates the dynamic changes of eccDNAs in rice across different growth stages and under nutrient stress conditions (nitrogen and phosphorus), focusing on their potential roles in plant growth, development, and adaptation to environmental stresses. Using Nanopore sequencing, RNA-seq, and ATAC-seq technologies combined with bioinformatics analysis, the authors systematically analyzed the size, origin, distribution, and categories of rice eccDNAs, as well as their association with gene expression, transposable elements (TEs), and potential roles in response to nitrogen and phosphorus deficiency. Overall, the paper highlights the significant role of rice eccDNAs in plant development and environmental response, providing new insights into the plasticity of plant genomes.

Response:

We thank this referee for highlighting our findings and the overall positive comments.

Despite the research value of this study, there are some issues that require further revision.

Q1. Line 125: Where are the evaluation results for these eccDNA identification tools (CIDER-Seq2, ECCsplorer, ecc_caller, and ecc_finder)? In particular, regarding CIDER-Seq2, can its input dataset be compatible with the Nanopore sequencing data format used in this study? In addition, I noticed another published software for eccDNA identification (FLED), which appears to offer higher efficiency and accuracy in eccDNA detection (<https://github.com/FuyuLi/FLED>). I recommend the authors take it into consideration.

Response:

We thank our referee for pointing out this ambiguity on line 125 and apologize for the over-compressed text that leads to this ambiguity.

According to the researchers referenced on CIDER-Seq and ecc_finder, CIDER-Seq2 (the updated version of CIDER-Seq) can identify eccDNAs from long-read sequencing data, particularly from PacBio long reads. However, it does not consider the bias introduced by genomic repeats, which are prevalent in the genomes of plants, including rice. ECCsplorer and ecc_caller focus on eccDNA identification from short-read sequencing but cannot process Nanopore sequencing data. We further refined this in our revised manuscript (lines 131-138), stating: “Instead of using the advanced tools of CIDER-Seq2 (Reference 24), FLED (New Reference 25), and CReSIL (New Reference 26), we opted for ecc_finder (Reference 27) for eccDNA identification in rice. This decision was based on ecc_finder’s precise bioinformatics framework, which accounts for genomic repeats, applies size filtering, and optimizes redundancy for eccDNA characterization. Moreover, ecc_finder has shown excellent performance with long-read Nanopore sequencing data in plants (Reference 17, 23, 27).”

We appreciate the reviewer’s recommendation regarding FLED.

FLED is a newer tool for eccDNA detection that utilizes long-read Nanopore sequencing data and employs bioinformatics strategies differing from those of ecc_finder and eccDNA_RCA_Nanopore. We conducted identification tests using FLED on our samples and further characterized the *ecGenes* from its outputs. Our results show that FLED detects more eccDNAs and *ecGenes* than ecc_finder (see bar charts below), aligning with findings from the FLED publication. However, the chromosomal distribution of FLED’s outputs exhibits a high level of redundancy (see karyoplots). As recently reported, this redundancy often arises from the identification of similar fragments originating from a single eccDNA, which skews detection efficiency, particularly in eccDNA-enriched experimental methods (see reference 43 in the revised manuscript). Thus, while FLED demonstrates superior detection capability over ecc_finder, its effective application to complex plant genomes requires a custom script to reduce redundancy before downstream analysis.

Meanwhile, we performed statistical analysis (see the histograms and statistic table below). The smallest size of eccDNA from FLED is around 40 bp, while the largest size is around 27 kb. We noticed that FLED doesn't exclude the eccDNAs smaller than 200 bp and the largest eccDNAs detected by FLED for each treatment were smaller than those identified by ecc_finder (37 kb). We did size distribution (< 6 kb were shown) on FLED's outputs, which was similar to that observed by ecc_finder, but with eccDNAs < 200 bp. Although there is a lack of custom script on reducing redundancy, Venn analysis on *ecGenes* from both FLED and ecc_finder has still been done for a further comparison. We found that an average of 80.0% of *ecGenes* from ecc_finder were detected by FLED. However, this proportion is much less than that in eccDNA_RCA_Nanopore detection, of which an average of 95.5% *ecGenes* from ecc_finder were also detected by eccDNA_RCA_Nanopore (line 523). Together these findings suggest that most, if not all the eccDNAs considered in our analysis were validated by two other methods.

	median (bp)	min (bp)	max (bp)	average (bp)	mode (bp)
all Control	496	36	29344	965	170
all Low N	381	40	25836	784	180
all Low P	615	33	27385	1109	182

We were also surprised to notice that FLED showed low efficiency or unbalanced detection on the *Ctrl_D7*, *Ctrl_D14*, *LP_D7*, and *LP_D14* samples compared to other samples. Both this situation and the lack of optimization on redundancy further led to an error during counting matrix normalization for short-term LP treatments, hindering our efforts to characterize exclusive *ecGenes*.

Considering the differences in FLED’s bioinformatics framework regarding size selection and redundancy optimization compared to *ecc_finder*, along with *ecc_finder*’s successful performance—particularly in plant eccDNA detection, as highlighted by previous reports and our findings—we decided to keep our analysis based on *ecc_finder* results in our manuscript. Additionally, to clarify information about various eccDNA detection tools, we adjusted our manuscript on lines 131-138 and revised reference 25 to provide a more accurate citation for FLED. We also clarified more details on eccDNA identification using *ecc_finder* in material and methods section (lines 886-893). We appreciate the referee’s understanding of our decision.

Q2. Line 147: Given that the average size of eccDNA is 500 bp, is it reasonable to merge sequences with an overlap of more than 200 bp? If the overlapping regions are merely shared sequences without representing identical functional regions, over-merging could lead to misinterpretations. Relying solely on overlapping to merge could result in the loss or misjudgment of information. In certain cases, eccDNAs may show significant variations due to transposable element activity or DNA rearrangement, and these variants might exhibit long overlaps. If merging is entirely based on a 200 bp overlap, some meaningful variations or diversity in fragments might be overlooked. The discussion should address the potential problems and limitations of this merging approach.

Response:

We fully acknowledge the referee’s concerns and want to clarify our merging process.

In our revised manuscript (lines 154-157), we refined that “during eccDNA identification with *ecc_finder* in each sample, redundancy is already optimized, and eccDNA regions enable the assignment of unique IDs. However, when generating IDs across all samples, certain genomic regions—particularly those with high eccDNA density—can lead to artificial duplication of IDs”. This occurs when eccDNAs from the same genomic region, which should

share a single ID, are instead assigned different IDs because they were identified in separate samples. To resolve this issue, we developed a merging process specifically designed to minimize ID duplication and ensure more accurate labeling across samples.

We analyzed the eccDNA size distribution and other statistical parameters in both control and treated sample outputs (Supplementary Figure 4) to determine an appropriate overlap length for reducing ID duplication. Given that the mode size of eccDNA reads is approximately 210 bp and the median size is around 340 bp, we established a threshold of 200 bp for merging overlapping eccDNAs into a unique ID across all samples (revised Supplementary Figure 5a). This approach significantly reduced the number of eccDNA IDs, enabling more rigorous downstream analysis, particularly in calculating proportions.

To ensure the accuracy of our merging process, we carefully validated the results by intersecting unmerged and merged data to check for any unexpected merging or over-merging of eccDNA IDs. As shown in the bar chart below, 61.2% of eccDNA IDs correspond to unique eccDNAs with no overlaps exceeding 200 bp, while 23.0% contain two eccDNAs, and 8.2% contain three. Only 7% of IDs include four or more eccDNAs, indicating that potential over-merging affects only a small fraction of eccDNAs.

We also validated our approach by examining the eccDNA IDs that merged the highest number of eccDNAs across samples. Our analysis confirmed that all these regions genuinely intersect and can be accurately assigned a single eccDNA ID (see the table within the bar chart), reinforcing the reliability of our merging strategy.

Following our merging process, we identified a total of 96,757 eccDNA IDs across all samples (Supplementary Figure 5b). Importantly, this process preserved the full diversity of eccDNAs, ensuring that the original loci from ecc_finder's output remained unchanged. The BED file containing the detailed loci of each eccDNA, along with their assigned merged IDs, was then used for downstream analysis of genes, transposable elements (TEs), and non-coding RNAs (ncRNAs) to calculate their proportions, as shown in the exit box of Supplementary Figure 5a after revision. We noticed the lack of an explanation of the merging process in the previous version of our manuscript, and a revised text is included in this new version of our paper (lines 154-162), also in the legend and caption of Supplementary Figure 5a.

Q3. Line 151: The definition and limitations of ecGenes. When ecGenes contain complete genes, how can it be distinguished whether gene expression originates from genomic amplification or eccDNA amplification? When ecGenes include only partial fragments of genes, how can the causal relationship between changes in gene expression and ecGenes be determined?

Response:

We fully understand the referee's concerns and acknowledge that addressing these questions requires further research, not only in this study but across all published plant eccDNA studies. While our revised manuscript outlines potential research directions for analyzing *ecGene* transcripts and functions (lines 772-779 and 788-792), current technologies in rice do not allow us to distinguish whether a transcript originates from genomic loci or from an amplified full-length gene-overlapping eccDNA (lines 788-800). Similarly, due to technological limitations, establishing a direct causal relationship between partial eccDNA-genes and changes in gene expression remains challenging (lines 774-777). In the next phase of our eccDNA research, we aim to develop specialized tools and molecular strategies to reliably determine the link between gene expression and *ecGene* accumulation.

Q4. Line 185: Regarding the classification of differentially enriched ecGenes as "exclusive", some limitations should be addressed: 1) Insufficient sequencing depth: If sequencing depth is inadequate, especially for low-abundance eccDNAs, the expression of certain genes may not be fully captured. This might lead to false negatives for genes that should exhibit significant differences. Conversely, genes with fewer reads in one treatment group may be mistakenly identified as differentially expressed due to sequencing errors. 2) Lack of biological context and functional validation: While using q-values and biological replicates to identify differentially expressed genes is reasonable, without additional functional validation or biological context, these genes might only represent statistical differences without real biological significance. These all need to be explained and clarified in detail during the discussion.

Response:

We agree with the referee that sequencing depth could influence the identification of differentially abundant eccDNAs. Therefore, after consulting with experts in the eccDNA field, who suggested that sequencing coverage should be equivalent to 15X genome size, we performed over 20X genome coverage on each sample to achieve reasonable sequencing depth for eccDNA identification. Moreover, a recent study on the comparative analysis of eccDNA identification tools reveals that when sequencing depth reaches around 20X, the accuracy of eccDNA detection from *ecc_finder* becomes stable and does not cause unexpected redundancy (reference 43).

After performing differential analysis on the gene-overlapped eccDNA counting matrix using the GMPR + EMDomics pipeline, following the framework's requirements, we identified *ecGenes* that were significantly abundant in one condition while absent in the other (exclusive *ecGenes*). We further clarified our material and methods on lines 900-912. In the previous version of our manuscript, we conducted inverse PCR in multiple cases: one widely existing *ecGene* (Supplementary Figure 8d-f), two typical exclusive *ecGenes* (Figure 2e-f; Figure 3e-f),

and one differential *ecGene* (Figure 4d-f) as experimental validations on our identification of *ecGenes*. We noticed the lack of a detailed discussion on the biological significance and additional function analysis of exclusive *ecGenes*, to clarify more on these, we further emphasized the necessity for future validation studies to explore the functional roles of *ecGenes* and elaborated on this discussion in lines 679-685, 699-701, 736-739 and 756-757 in our revised manuscript.

Q5. Line 318: About the issue of the relationship between RNA-seq and ecGenes. Although the RNA-seq analysis revealed a correlation between some ecGenes and chromosomal gene expression, this remains an association rather than a causative relationship. For example, while the authors note that certain ecGenes are associated with upregulated or downregulated chromosomal genes, it remains unclear whether these ecGenes directly participate in gene expression regulation. Further functional validation experiments are needed to establish causality.

Response:

Thanks for the concerns. We fully understand our referee.

We analyzed association between exclusive *ecGenes* and differentially expressed chromosomal genes to understand their complex potential functions further. We modified our results (lines 342, 358, 431, 443), discussion (lines 761 and 780), and captions on Supplementary Figure 11 and 14 to ensure not making any overstatement on the association between *ecGene* accumulation and chromosomal gene expression. We cautiously discussed our association analysis results and revised line 768-770 in our manuscript to avoid misleading the readers. Moreover, to emphasize the need for innovative strategies for exploring the multiple mechanisms of *ecGenes* in rice, we revised our discussion by providing detailed hypotheses, for instance, the possible role of a novel siRNA regulation and several prospective related research strategies (lines 772-779 and lines 788-792).

Q6. For the Nanopore sequencing and ATAC-seq data, it is recommended to provide more detailed statistics on sequencing depth and library saturation analysis, including a clear explanation of how biological replicates were managed for each treatment to minimize noise and validate the robustness of the observed differences. While the authors mention LongQC in the Methods section, additional details on the actual average coverage depth, duplication rates, and replicate correlations should be included in the Supplementary Data.

Response:

Thanks for the recommendation from our referee.

We revised and provided new supplementary data on Nanopore sequencing (Supplementary Data 1) and ATAC-seq (Supplementary Data 15) to include the necessary statistical information on quality control (QC) and sequencing depth. We also updated our materials and methods on lines 826-828, 866-870, 880-884, 992-994, and 1010-1017 to better explain the strict management of biological replicates, library preparation, and QC analysis. To support the differences observed during optimal growth and under nutrient limitations, we elaborated on the nutritional treatment settings and sample collections on lines 111-118, as well

as the eccDNA identification process for both Nanopore sequencing (lines 125-129) and ATAC-seq (lines 570-582).

Q7. When identifying differentially enriched eccDNAs, particularly differential ecGenes, the authors used GMPR normalization and EMDomics, which is conceptually feasible. However, given that zero counts of eccDNA are very common, have the authors considered other zero-inflated models (e.g., ZINB, etc.) for comparison? Alternatively, can additional sensitivity analyses be conducted from a methodological standpoint? Presenting the consistency of differential results under different normalization/statistical models would be more convincing.

Response:

Thanks for the suggestion. We tested several pipelines of normalization and differential analysis for *ecGene* research based on the zero-inflation model shown in the data matrix:

a) Normalization through GMPR + differential analysis through EMDomics (as the analysis in our manuscript);

b) Normalization through tweedEseq + differential analysis through tweedEseq; tweedEseq^[1] is a tool based on *Poisson-Tweedie*, the flexible count data model, to fit a wide diversity of expression profiles arising from sequencing experiments, including zero-inflation. According to its framework, it provides unique normalization and statistical methods that can handle the situation of zero inflation during differential analysis.

We performed tweedEseq on LN and LP treatment data to validate the difference in *ecGenes* abundance under different nutrient limitations.

c) Normalization through tweedEseq + differential analysis through EMDomics; Considering the possible bias caused by normalization methods, we further tested the combination of tweedEseq normalization methods and EMDomics differential analysis to double-check our findings on differentially enriched and exclusive *ecGenes*.

We conducted a Venn analysis on differentially enriched and exclusive *ecGenes* for LN and LP treatments to determine whether the various statistical approaches yielded consistent or differing results. We observed highly consistent outcomes across all tested methods, as shown in the Venn diagrams below. Our findings regarding differentially enriched and exclusive *ecGenes* are consistent since all methods identified the same enriched and exclusive *ecGenes* (over 90%). In contrast to normalization plus differential analysis using tweedEseq (twee_twee in purple) and normalization through tweedEseq combined with differential analysis via EMDomics (twee_EMD in blue), we ultimately selected normalization through GMPR and differential analysis through the EMDomics pipeline (GMPR_EMD in green) due to its high accuracy and efficiency for detecting both differentially enriched and exclusive *ecGenes*.

[1] Esnaola, M., Puig, P., Gonzalez, D., Castelo, R., & Gonzalez, J. R. (2013). A flexible count data model to fit the wide diversity of expression profiles arising from extensively replicated RNA-seq experiments. *BMC bioinformatics*, 14, 254.

Q8. The authors performed GO enrichment analysis for the differential ecGenes and used REVIGO to consolidate the results. However, if only part of a gene (e.g., partial CDS/UTR) is covered, it is not guaranteed that the function of that segment is the same as the full-length gene. It is recommended that the authors make a clearer distinction between “partial gene overlap” vs. “full-length overlap” in the Results and Discussion sections to avoid over-interpretation.

Response:

We fully understand our referee’s concerns. Therefore, we revised these important points in the new version of our manuscript.

According to our analysis, the counts of full-length *ecGenes* in the exclusive *ecGenes* we characterized from different treatments varied from 2 to 24, which is insufficient for generating separate GO results. Meanwhile, separating the full-length *ecGenes* from the exclusive *ecGenes* didn’t change the categories identified in GO or REVIGO analysis. Considering the potential functions of modifying gene expression of both full-length and partial *ecGenes*, we finally decided to perform the GO and REVIGO analysis for all exclusive *ecGenes* together to address a more comprehensive overview of the role played by them in rice genome to respond to nutrient limitations.

To avoid confusion, we further clarify the details regarding full-length *ecGenes* in our manuscript on lines 212, 228, 241, 293, 312, 374, and 390 in the results section, as well as on lines 681-682 in the discussion section. These revisions help prevent over-interpreting our findings. Additionally, we provide new Supplementary Data 3, 7, and 11 to present all gene IDs of full-length exclusive *ecGenes* during optimal growth and under nutrient limitations.

Q9. The article discusses in detail the potential activity of MITEs such as *Stowaway* and *Kiddo*, as well as LTRs, which is a novel highlight of this study. It is recommended that the authors include more explanation in discussion on the criteria for judging TE activity. For example, is there any evidence indicating that these elements indeed transpose or affect the expression of nearby genes under stress (e.g., changes in RNA-seq expression upstream or downstream of insertion sites)? The terms “active” or “potentially active” should be used more cautiously.

Response:

Following the referee’s suggestion, we revised our results (lines 490-493) and discussion on transposons and repeat units present in eccDNAs (lines 652-658 and 667-676) to clarify the potential activity more cautiously. According to our analysis, *Stowaway* units classified as MITEs can originate as circular molecules during both optimal growth and nutritional stress (Figure 5e-g and Supplementary Figure 16). Conversely, *Kiddo* units form circular molecules specifically in response to nutrient limitations (Supplementary Figure 15d-f). The formation of *Stowaway-ecRepeatUnits* and *Kiddo-ecRepeatUnits* has been proposed as evidence of the potential activity of MITEs in rice (lines 654-658 and 667-670). Meanwhile, we noted the limitations of future functional studies on existing or newly inserted circular units (lines 671-673). We hypothesized on DNA-methylation-related mechanisms based on several reference studies (lines 674-676). To be more precise, we refined our terminology to “potentially active” in both results and discussion.

Q10. Some minor errors should be corrected, e.g., in *Line 153*, it should be written as 30.3% instead of simply 30%; similarly, in *Line 157*, it should be written as 8.0% instead of simply 8%. The entire text should be carefully proofread.

Response:

We corrected these errors (lines 166-171, lines 459-465) and proofread the manuscript in its entirety.

In conclusion, this study offers valuable insights into the potential roles of rice eccDNAs in plant development, particularly in environmental stress response, and proposes new hypotheses and research directions. However, the manuscript has some weaknesses in data analysis depth and exploration of biological mechanisms. To enhance the quality of the article, I recommend the authors address the aforementioned comments, providing additional reliable evidence.

Response:

Thanks again for highlighting our valuable insights and addressing the concerns. We hope our referee is satisfied with our responses and our revised manuscript.

Response to Referee #4:

The authors have adequately addressed the reviewers’ concerns. The revised manuscript has undergone significant improvements, which is now a valuable contribution to eccDNA research. However, I have two comments regarding the selection of analysis tools.

Major issues:

1. In the *ecc_finder* paper [1], the authors state, “We have not tested *ecc_finder* on ATAC-seq data, but a recent method for this specific type of data has been described.” The specific algorithm for ATAC-seq data analysis is *Circle_finder*[2]. Why did the authors opt to use *ecc_finder* rather than *Circle_finder* for the ATAC-seq data analysis presented in Figure 7?

References:

[1] Zhang, P., Peng, H., Llauro, C., Bucher, E. & Mirouze, M. *ecc_finder: A Robust and Accurate Tool for Detecting Extrachromosomal Circular DNA From Sequencing Data*. *Front. Plant Sci.* 12, 743742 (2021)

[2] Kumar, P. *et al.* ATAC-seq identifies thousands of extrachromosomal circular DNA in cancer and cell lines. *Sci. Adv.* 6, eaba2489 (2020).

Response:

We agree with the reviewer that this is an important question.

According to the report by Kumar *et al.*, circular molecules were preserved during the library preparation of ATAC-seq. Split-read identification can further distinguish them in the short-read sequencing data (reference 44). The framework designed in the short-read mode of *ecc_finder* enables the detection of eccDNAs from ATAC-seq data (reference 27). Moreover, recently, Kang *et al.* published results demonstrating the performance of the ATAC-seq - *ecc_finder* pipeline in identifying eccDNA molecules (reference 45). Compared to *Circle_finder*, *ecc_finder* incorporates essential measures to account for genomic repeats and reduce false positives through its precise analytical framework, demonstrating strong performance in detecting plant eccDNA from short-read sequencing data (reference 17, reference 27, reference 43). Given these advantages, we selected *ecc_finder* for our ATAC-seq validation. We recognize that our previous manuscript lacked a detailed explanation of this choice, and we have now provided further clarification in our revised manuscript on lines 570-578 and 579-582 (Figure 7 section in Results) and line 1016 (ATAC-seq and eccDNA validation section in Materials and Methods).

reference 17:

Zhuang, J., Zhang, Y., Zhou, C., Fan, D., Huang, T., Feng, Q., Lu, Y., Zhao, Y., Zhao, Q., Han, B., & Lu, T. (2024). Dynamics of extrachromosomal circular DNA in rice. *Nature communications*, 15(1), 2413.

reference 27:

Zhang, P., Peng, H., Llauro, C., Bucher, E., & Mirouze, M. (2021). *ecc_finder: A Robust and Accurate Tool for Detecting Extrachromosomal Circular DNA From Sequencing Data*. *Frontiers in plant science*, 12, 743742.

reference 43:

Gao, X., Liu, K., Luo, S., Tang, M., Liu, N., Jiang, C., Fang, J., Li, S., Hou, Y., Guo, C., & Qu, K. (2024). Comparative analysis of methodologies for detecting extrachromosomal circular DNA. *Nature communications*, 15(1), 9208.

reference 44:

Kumar, P., Kiran, S., Saha, S., Su, Z., Paulsen, T., Chatrath, A., Shibata, Y., Shibata, E., & Dutta, A. (2020). ATAC-seq identifies thousands of extrachromosomal circular DNA in cancer and cell lines. *Science advances*, 6(20), eaba2489.

reference 45:

Kang, J., Dai, Y., Li, J., Fan, H., & Zhao, Z. (2023). Investigating cellular heterogeneity at the single-cell level by the flexible and mobile extrachromosomal circular DNA. *Computational and structural biotechnology journal*, 21, 1115–1121.

2. *The authors applied eccDNA_RCA_nanopore to identify multiple-fragment eccDNAs. However, a recent benchmark study[3] indicates that CReSIL[4] outperforms eccDNA_RCA_nanopore in identifying eccDNAs from Nanopore data. Additionally, the identification results using eccDNA_RCA_nanopore exhibit redundancies. Therefore, I recommend that the authors consider using CReSIL for the analysis of multiple-fragment eccDNAs to validate their results.*

References:

[3] Gao, X. et al. Comparative analysis of methodologies for detecting extrachromosomal circular DNA. *Nat. Commun.* 15, 9208 (2024).

[4] Wanchai, V. et al. CReSIL: accurate identification of extrachromosomal circular DNA from long-read sequences. *Brief Bioinform.* 23 (2022).

Addressing these points will strengthen the manuscript, ensuring that it provides a comprehensive, accurate, and clear resource for readers interested in eccDNA research.

Response:

We appreciate our referee's recommendation regarding CReSIL.

CReSIL is a tool for eccDNA identification from Nanopore sequencing data, based on a reconstruction and *de novo* assembly framework, which is very different from ecc_finder and eccDNA_RCA_Nanopore. We tested CReSIL on our samples and compared the multiple-fragment eccDNAs (MF-eccDNAs) identified by this tool to those obtained using eccDNA_RCA_Nanopore. We found that the largest number of fragments in the MF-eccDNAs detected by CReSIL was 10, consistent with our results using eccDNA_RCA_Nanopore (see the bar chart below). In addition, 99.8% of eccDNAs detected by CReSIL were single fragment eccDNAs, which is also consistent with the proportion from eccDNA_RCA_Nanopore (99.7%, on line 526).

Upon further analysis of the four typical sets of MF-eccDNAs, we found that CReSIL only detected Os01g0791033-MF-eccDNAs in several treatments, while this type of MF-eccDNAs was present in all treatments when using eccDNA_RCA_Nanopore. CReSIL can detect Os04g0343050-MF-eccDNAs in one control sample. However, CReSIL failed to detect Os04g0343050-MF-eccDNAs in any of the treatment samples, despite our experimental identification of this MF-eccDNA using inverse PCR in the samples of LN_D7 and LP_D14. Additionally, CReSIL does not detect Os05g0372300-MF-eccDNAs or Os12g0423313-MF-eccDNAs. Therefore, we conclude that eccDNA_RCA_Nanopore is more effective in identifying MF-eccDNAs in plants.

For eccDNA RCA Nanopore results, we took several steps to reduce redundancy for MF-eccDNAs during our analysis: 1) identifying the core gene overlapping with MF-eccDNAs; 2) grouping the MF-eccDNAs into different sets based on the core gene; and 3) tracing specific regions characterized in single reads of particular sets of MF-eccDNAs to verify the variety and calculate the read numbers for this specific set of MF-eccDNAs. Through these steps, we proposed four typical sets of MF-eccDNAs in our manuscript and conducted further analysis on them.

We clarified our results on lines 529-531 and material and methods section on lines 968-972, revised reference 43 to provide more detailed information, and explained how we addressed the redundancy. We also revised the legend and caption of Figures 6a-b to prevent any misunderstanding due to redundancy. Additionally, we added sample names and read IDs in our supplementary data 14 (previously supplementary data 11) to provide more details as needed. Meanwhile, to clarify information about various eccDNA detection tools, we adjusted our manuscript on lines 131-138 and revised reference 26 to provide a more accurate citation of CReSIL. Considering all these factors, we ultimately decided to maintain our findings based on eccDNA_RCA_Nanopore for MF-eccDNA analysis. We appreciate our referee's understanding of our decision.

reference 26:

Wanchai, V., Jenjaroenpun, P., Leangapichart, T., Arrey, G., Burnham, C. M., Tümmeler, M. C., Delgado-Calle, J., Regenber, B., & Nookaew, I. (2022). CReSIL: accurate identification of extrachromosomal circular DNA from long-read sequences. *Briefings in bioinformatics*, 23(6), bbac422.

reference 43:

Gao, X., Liu, K., Luo, S., Tang, M., Liu, N., Jiang, C., Fang, J., Li, S., Hou, Y., Guo, C., & Qu, K. (2024). Comparative analysis of methodologies for detecting extrachromosomal circular DNA. *Nature communications*, 15(1), 9208.

Response to the Editor and Reviewer Comments

We thank the reviewers and the editor for their thoughtful comments and suggestions on our manuscript. Your insights have been invaluable in highlighting areas for improvement and enhancing the overall quality of our manuscript. We appreciate the time and effort dedicated to reviewing our manuscript.

As requested by the reviewers, the manuscript has been proofread, and grammar and spelling have been thoroughly revised to correct all errors.

Below, we provide a point-by-point response to the reviewer's comments.

Response to Reviewer #3:

Overall Recommendation:

Accept after minor revisions (English language polishing)

Major Concerns Addressed:

The authors have made satisfactory revisions in response to the previous round of review comments, and the manuscript now aligns with the scientific standards of Nature Communications. However, several issues in English writing persist and require careful polishing to ensure clarity and professionalism.

Response:

We thank the reviewer for the valuable comments throughout the peer review process. The valuable guidance helped us significantly improve the clarity and rigor of our study.

Specific Revisions Required:

There are several inconsistencies in subject-verb agreement throughout the manuscript. For instance, at line 163, "of which 267 covered full-length gene" should be revised to "of which 267 covered full-length genes."; similarly, at line 207, 223, 236, 288, 306, 367, 383 etc. Additionally, minor spelling errors were noted. One example is at line 670, where "segment" was misspelled as "sagment". These and other similar errors should be carefully revised throughout the text. A professional proofreading service or a native English-speaking co-author should review the manuscript to address these issues.

Response:

We thank the reviewer for highlighting the inconsistencies in grammar and spelling. This version of the manuscript has been thoroughly revised to enhance the English language. The corrections include, among others, the misspellings and typos found on lines 163, 207, 223, 236, 288, 306, 367, 383, 437, and 677. Thank you for your detailed and valuable revision.

Response to Reviewer #4:

My concerns are adequately addressed in the revised manuscript. I have no further questions.

Response:

We appreciate the reviewer's comments and suggestions during the peer review process, which significantly aided us in enhancing the quality of our work.